# When complex neuronal structures may not matter

**Adriane G Otopalik[1]\*, Alexander C Sutton[1†], Matthew Banghart[2‡], Eve Marder[1]\***

[1]Volen Center, Biology Department, Brandeis University, Waltham, United States; [2]Department of Neurobiology, Harvard Medical School, Boston, United States

**Abstract** Much work has explored animal-to-animal variability and compensation in ion channel expression. Yet, little is known regarding the physiological consequences of morphological variability. We quantify animal-to-animal variability in cable lengths (CV = 0.4) and branching patterns in the Gastric Mill (GM) neuron, an identified neuron type with highly-conserved physiological properties in the crustacean stomatogastric ganglion (STG) of *Cancer borealis.* We examined passive GM electrotonic structure by measuring the amplitudes and apparent reversal potentials ($E_{rev}$s) of inhibitory responses evoked with focal glutamate photo-uncaging in the presence of TTX. Apparent $E_{rev}$s were relatively invariant across sites (mean CV $\pm$ SD = 0.04 $\pm$ 0.01; 7–20 sites in each of 10 neurons), which ranged between 100–800 μm from the somatic recording site. Thus, GM neurons are remarkably electrotonically compact (estimated $\lambda$ > 1.5 mm). Electrotonically compact structures, in consort with graded transmission, provide an elegant solution to observed morphological variability in the STG.

**\*For correspondence:** aotopali@ brandeis.edu (AGO); marder@ brandeis.edu (EM)

**Present address:** [†]Boston Consulting Group, Boston, Massachusetts, United States; [‡]Neurobiology Section, Division of Biological Sciences, University of California San Diego, San Diego, United States

## Introduction

Neuronal circuits can generate stable output despite variable underlying parameters across animals (*Marder and Goaillard, 2006*; *Marder, 2011*). Work in invertebrate central pattern-generating circuits shows that circuit function can be maintained across animals despite variable synaptic, intrinsic, and modulator-induced currents in identified constituent neurons (*Prinz et al., 2004*; *Schulz et al., 2006*; *Marder and Goaillard, 2006*; *Goaillard et al., 2009*; *Norris et al., 2011*; *Marder, 2011*; *Roffman et al., 2012*; *Williams et al., 2013*; *Gutierrez et al., 2013*; *Rodriguez et al., 2013*). This previous work has revealed the principle that neurons compensate for variable ionic conductances at the circuit (*Grashow et al., 2010*) and single-neuron levels (*Tobin et al., 2009*; *Ball et al., 2010*; *O'Leary et al., 2013*, *2014*).

The unique physiology of any given neuron is a consequence of its palette of ionic conductances, and its morphology (*Mainen and Sejnowski, 1996*; *Vetter et al., 2001*). A neuron's distributed, geometric cable properties: the length, diameter, taper, and branching of its neurites, shape passive current flow and voltage propagation (*Rall, 1959*, *1977*; *Goldstein and Rall, 1974*). This resulting electrotonic structure plays a central role in determining whether voltage signals arising at disparate sites across the dendritic tree are integrated or segregated (*Rall, 1959*, *1967*, *1969a*, *1969b*; *Rall and Rinzel, 1973*, *1974*; *Goldstein and Rall, 1974*; *Agmon-Snir and Segev, 1993*; *Vetter et al., 2001*). In this way, ion channel expression, when superimposed on morphology, gives rise to neuronal physiology, input-output computations, and circuit-level function (for reviews, see *Koch and Segev, 2000*; *London and Häusser, 2005*).

In the present study, we investigated the physiological consequences of animal-to-animal variability in neuronal morphology in the crab stomatogastric ganglion (STG), a central pattern-generating circuit composed of 26–27 neurons. The identified neuron types of the STG exhibit highly conserved physiological waveforms and circuit-level functions (*Harris-Warrick et al., 1992*), despite their

complex and variable morphologies across animals (*Wilensky et al., 2003*; *Baldwin and Graubard, 1995*; *Bucher et al., 2007*; *Goeritz et al., 2013*; *Otopalik et al., 2017*). STG neurons display large somata (50–100 μm in diameter) and primary neurites that ramify throughout the STG neuropil (*Selverston et al., 1976*; *King, 1976a*, *1976b*; *Baldwin and Graubard, 1995*; *Kilman and Marder, 1996*).

Synaptic transmission between neurons is predominantly graded, inhibitory cholinergic and gluta-matergic transmission (*Eisen and Marder, 1982*; *Marder and Eisen, 1984*; *Maynard and Walton, 1975*; *Graubard et al., 1980*; *Manor et al., 1997*, *1999*). Synaptic sites are sparsely distributed throughout finer processes in the neuropil region (*King, 1976a*, *1976b*). There are no synapses on somata in these neurons (*King, 1976a*, *1976b*) and spike initiation zones are located in the periphery where axons exit the ganglion (*Raper, 1979*; *Miller, 1980*). Thus, synaptic integration and release occurs predominantly in the neuropil. In individual STG neurons, pre- and post-synaptic sites are often tightly apposed on the same neurites (*King, 1976a*, *1976b*). This juxtaposition of synaptic input and output suggests that current will flow in all directions across the neurite tree, centripetally and centrifugally, in the intact circuit, and allow for integration of voltage signals arising from disparate loci on the neurite tree, should the neuron be sufficiently electrotonically compact.

Previous investigation of electrotonic structure in STG neurons employed simultaneous electrophysiological recordings at the soma and primary neurite, wherein electrodes were separated by several hundred microns (*Miller, 1980*; *Golowasch and Marder, 1992*). These recordings showed low-pass filtering of high-frequency voltage events across the primary neurite, as is typically imposed by the membrane capacitance (*Rall, 1977*). In contrast, slow voltage oscillations were subject to less electrotonic decrement. This frequency-dependent decay of electrical signals was consistent with the findings of contemporary theoretical and experimental works (*Rall, 1977*; *Johnston and Brown, 1983*; *Spruston et al., 1993*, *1994*; *Jaffe and Carnevale, 1999*). This early characterization of electrotonic structure in the STG may have been a satisfactory description at the time, given that STG circuit function is mediated predominantly by graded transmission (*Eisen and Marder, 1982*; *Marder and Eisen, 1984*; *Maynard and Walton, 1975*; *Graubard et al., 1980*; *Manor et al., 1997*, *1999*) and slow oscillations (*Graubard and Ross, 1985*; *Ross and Graubard, 1989*), and continues to oscillate in the absence of spikes (*Graubard, 1978*; *Raper, 1979*; *Graubard et al., 1983*; *Anderson and Barker, 1981*). However, these early experiments left the electrotonic properties of more distal sites and higher-order branches to the imagination.

Relevant voltage events must arise at more distal, finer processes, where pre- and post-synaptic connections are located (*King, 1976a*, *1976b*; *Kilman and Marder, 1996*). Thus, a full experimental characterization of the electrotonic structure of STG neurons requires electrophysiologically sampling numerous sites across the dendritic tree. Of course, recording at many sites, let alone on the tiniest of neurite processes, presents technical challenges that few have overcome in situ. In one STG neuron type, the gastric mill (GM) neuron, we have utilized focal photo-uncaging techniques in tandem with electrophysiology to examine propagation of voltage events evoked at processes that vary in size and distance from the somatic recording site. We present a surprising case wherein geometrical complexity and variability appear not to constrain passive physiology.

## Results

The stomatogastric ganglion (STG) of the crab *Cancer borealis* is composed of 26–27 neurons situated around a dense neuropil region, wherein each of the neurons branches extensively. This central-pattern-generating circuit mediates the coordinated rhythmic contractions of the animal's foregut. The gastric mill (GM) neuron is one of fourteen identified neuron types in the STG. There are typically four GM neurons in each animal. *Figure 1A* shows a schematic of one GM neuron and its axonal projections in the intact stomatogastric nervous system, as dissected for in vitro experiments. Filling the neuron with fluorescent dye reveals the complex morphology of the GM neuron in situ (*Figure 1B*). The GM neuron ramifies throughout the neuropil region and can be distinguished by multiple axons, projecting to and innervating extrinsic gastric muscles 1a, 1b, 2, 3a, and 3b via the anterior lateral (*aln*) and dorsal ventral (*dvn*) nerves (*Maynard and Dando, 1974*; *Selverston and Mulloney, 1974*; *Weimann et al., 1991*). GM neurons participate in the episodic gastric mill rhythm (*Hartline and Maynard, 1975*). When bursting (*Figure 1C*), GM activity evokes rhythmic contractions of its target muscles, resulting in grinding movements of the gastric mill ossicles and attached

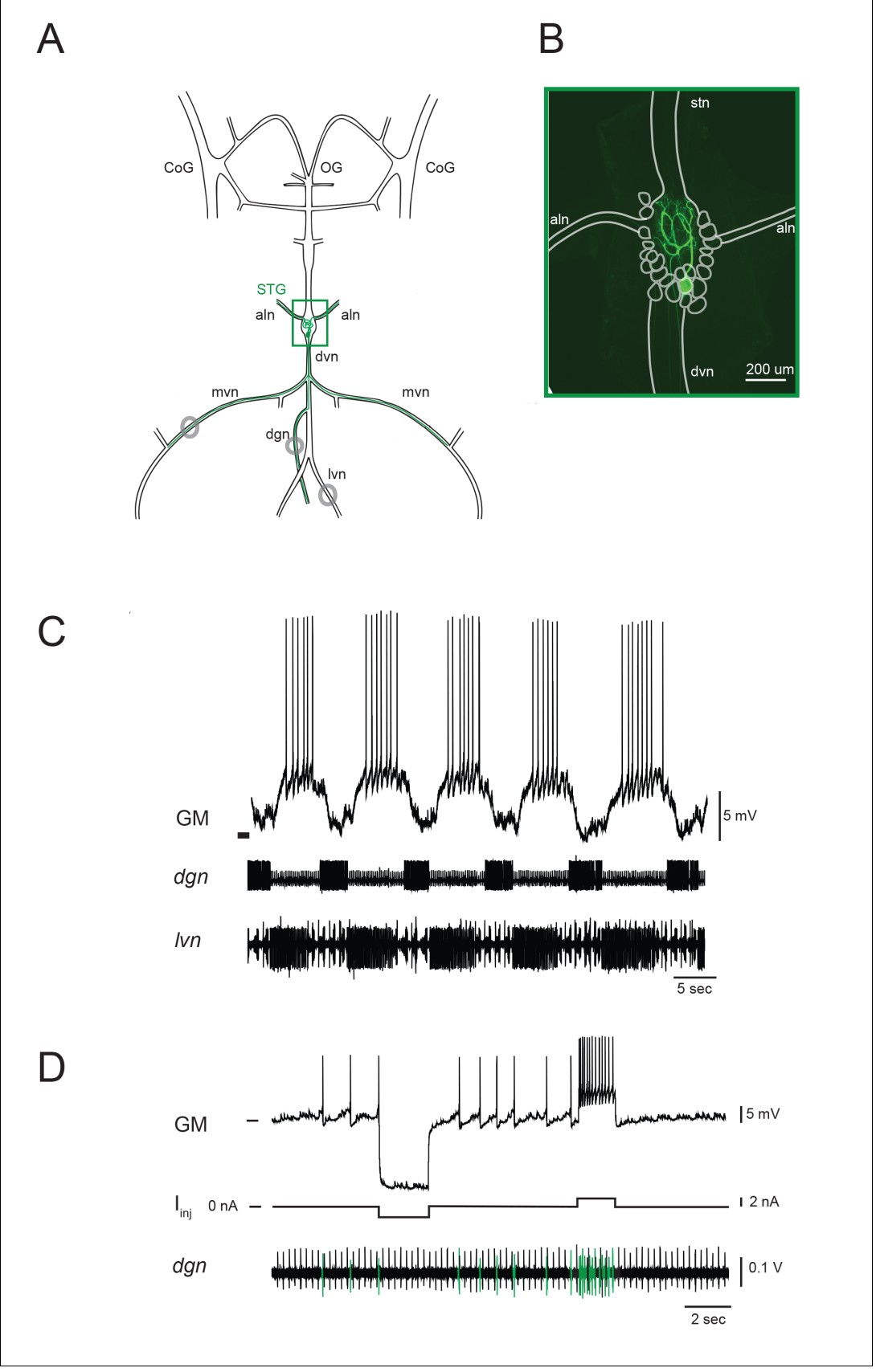

**Figure 1.** The stomatogastric nervous system (STNS) and identification of gastric mill (GM) neurons. (**A**) Schematic of an in vitro, isolated STNS with descending inputs intact (from bilateral commissural ganglia (CoGs) and esophogeal ganglion (OG)). The stomatogastric ganglion (STG; green box) contains identifiable motor neurons that project their axons onto specific muscle groups via the medial ventral nerve (*mvn*), anterior lateral nerves (*aln*), dorsal ventricular nerve (*dvn*) and lateral ventricular nerve (*lvn*), and dorsal gastric nerve (*dgn*). An example of the axonal projection path of a GM neuron is shown in green. Extracellular nerve recordings (locations indicated with gray circles) are utilized for physiological identification of GM neurons (as in **D**). (**B**) Example of an alexa488 dye-fill (green) of a GM neuron, with the STG, nerve projections, and other neurons outlined in white. (**C**) The GM neuron participates in a slow, episodic gastric mill rhythm that is highly conserved across animals. Top: an intracellular recording of GM with simultaneous extracellular nerve recordings of the *dgn* and *lvn* show concurrent, single-neuron and circuit-level activities. (**D**) GM neurons are identified physiologically by matching intracellular spike activity with extracellular nerve spike units in the *dgn*, which contains GM axonal projections (shown in green on *dgn*). Hyperpolarizing and depolarizing current pulses ($I_{inj}$) make this identification unambiguous. These identifications are verified with microscopy and identification of axons projecting bilaterally into the *alns,* as in **B**. In **C** and **D**, horizontal bars indicate an intracellular voltage of −50 mV.

teeth for internal chewing of food (*Russell, 1985*; *Heinzel, 1988*). GM neurons are unambiguously identified by matching their spiking activity with spiking units on extracellular nerves known to contain GM axonal projections (*Figure 1D*; *Maynard and Dando, 1974*).

## Variability in GM neuron morphology

We generated three-dimensional confocal stacks of Lucifer Yellow dye-fills of 14 GM neurons situated in the STGs of 14 different animals. *Figure 2A* shows six examples of maximum z-projection images generated from GM neuronal dye-fills. GM neurons exhibit large somata (with a mean diameter ± SD of 74 ± 13 µm) and expansive neurite branches that span the neuropil region. Non-axonal neurite branches spanned an average ellipsoid volume of $4.98 \pm 1.43 \times 10^6$ µm$^3$ (mean ± SD).

Using KNOSSOs software (freely available online at knossostool.org, see Materials and methods), we manually traced and generated 3-dimensional skeletal reconstructions of these neuronal dye-fills (skeletal reconstructions of the six neuronal dye-fills in *Figure 2A* are shown in *Figure 2B*; *Supplementary file 1*-Neuronal Structures Hoc files). From these skeletal reconstructions, we were able to measure total cable lengths, number of branch points, and soma-to-tip path lengths and tortuosities (excluding axons) using a suite of custom morphology analytical tools (see Materials and methods). These neurons present expansive structures, with a mean total cable length of 8840 ± 3678 µm (*Figure 2C*) and numerous branch points (mean number of branch points ± SD was 155 ± 126; *Figure 2D*). Notably, across animals, total cable lengths and branch point numbers varied by 40% of the mean (CV = 0.4 for both metrics). Note that these numbers are underestimates and smaller than reported in *Otopalik et al. (2017)*. This is due to the lower resolution used for reconstruction of this large set of neurons, resulting in loss of some of the smallest profiles. *Otopalik et al. (2017)*, reported similar animal-to-animal variability in these metrics across a smaller sample size.

We also measured all soma-to-tip paths within each neuron (*Figure 2E*). These path length distributions varied within and across neurons, with distances ranging between 200–1000 µm and a mean coefficient of variation of 0.81. The mean soma-to-tip path length was 450 ± 80 µm. Thus, if synaptic voltage events arising at the neurite tip are measured electrophysiologically at the soma, such signals would travel this distance, on average.

In *Figure 2A and B*, it is visually evident that GM neurons display tortuous three-dimensional structures. This is quantified in *Figure 2F*, which shows the tortuosities of soma-to-tip paths (path length/Euclidean distance from soma to tip location). For a given soma-to-tip path, a tortuosity of one suggests a minimal, Euclidean path from soma to tip, whereas tortuosities greater than one suggest winding paths that deviate from the minimal Euclidian distance (*Figure 2F*, right). Across GM neurons, tortuosity distributions vary. But, each neuron presents a broad tortuosity distribution and has a mean tortuosity that is modestly greater than one (*Figure 2E*; the mean across all neurons is 2.1 ± 0.5). Taken together, these data suggest that GM neurons are tortuous and expansive, and somewhat variable in their macroscopic morphology.

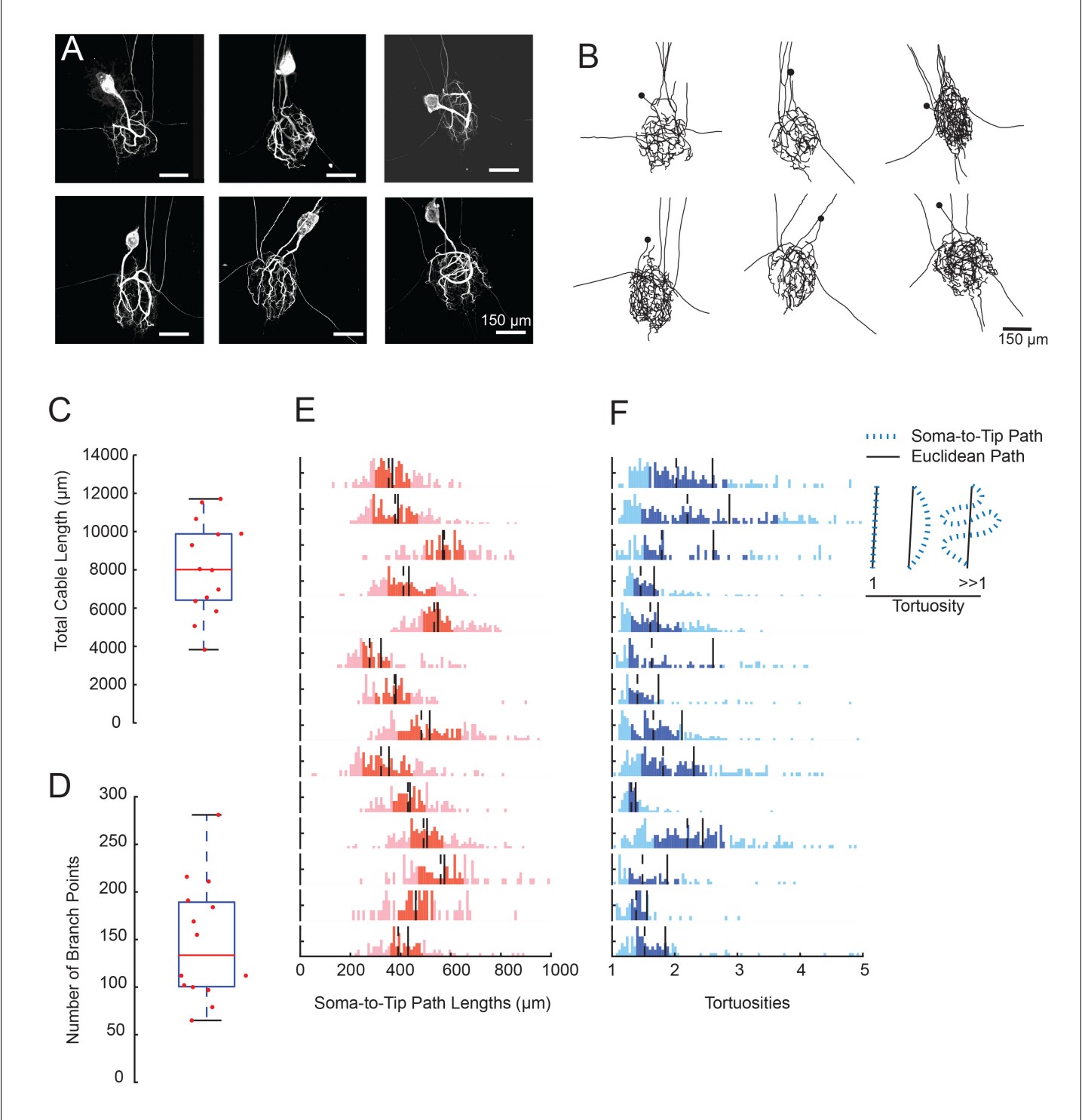

**Figure 2.** GM neurons exhibit expansive and complex morphologies. (**A**) Maximum z-projections of 3-dimensional confocal image stacks capturing Lucifer yellow dye-fills of six GM neurons (taken at 20x magnification). (**B**) Skeletal reconstructions of the six neurons shown in **A**, generated by manual tracing in KNOSSOS software and used for quantitative morphological analyses shown in **C–F**. All scale bars in **A** and **B** are 150 µm. (**C**) Boxplot of total cable lengths (excluding axonal projections; Mean + SD = 8112.6 + 2453.6 µm). (**D**) Boxplot of the total number of branch points (excluding axonal branch points; mean + sd = 148 + 63 branch points). For **C** and **D**, red line indicates the median, the blue box spans the 25th and 75th percentiles, and the whiskers span the range of data points not considered to be outliers. (**E**) Histograms show the distributions of paths from soma to terminating neurite for individual GM neurons (from top to bottom). (**F**) Histograms showing the tortuosities for soma-to-tip paths for individual GM neurons (from top to bottom). Tortuosity was calculated as the ratio of measured soma-to-tip path length (as in **E**) over the Euclidean distance from soma to tip. On

*Figure 2 continued on next page*

*Figure 2 continued*

right, diagram indicates the interpretation of tortuosity for a given path (blue) relative to the Euclidean distance (black). For E and F, the darker shaded region spans the 25th and 75th percentiles, solid lines indicate the mean, and dashed lines indicate the median. The y-axes are scaled to the maximum number of paths among bins within each neuron. Note the leftward bias in the tortuosity distributions (**F**). For C–F, metrics were calculated for N = 14 GM neurons in 14 different animals.

The functional consequences of morphological complexity and variability are best considered in light of how voltage signals arising from presynaptic inputs are integrated and transformed into spike patterns. Previous work suggests that pre- and post-synaptic connections are distributed throughout the finer process within the STG neuropil (*King, 1976b*; *Kilman and Marder, 1996*) and that spike initiation zones are located in the periphery, where the axons exit the neuropil (*Raper, 1979*; *Miller, 1980*). Thus, axon number, location, and branching patterns relative to each neuron's neurite tree may speak to how voltage signals arising from presynaptic inputs are integrated. GM neurons typically have between 3–5 axons (*Figure 3A*), with at least one projection to the *aln* and one projection to the *dvn* (this is consistent with previous findings; *Maynard and Dando, 1974*). In 2/14 neurons, only two axons were identified. This is likely due to an incomplete dye-fill or atypical branching of axonal projections beyond the microscope's field of view. Branch order distributions varied across neurons, as did axonal branch point orders (*Figure 3B*). As is illustrated in *Figure 3C–G*, in some GM neurons all axons projected from the same branch point (as in *Figure 3D*), whereas in other GM neurons each axon projected from a distinct subtree (as in *Figure 3G*). Taken together, variable axon numbers, locations, and branching patterns suggest that each GM neuron differentially integrates voltage signals arising from varying proportions of lower and higher branch orders.

## Variable glutamate responses across the neuronal structure

One might expect that complex and variable neuronal morphologies would yield complex and variable electrotonic structures. To probe the distributed cable properties of GM neurons, we employed a custom-built microscope to focally photo-uncage MNI-glutamate (Tocris) with ultraviolet light at many sites across the GM neuronal structure (in TTX to attenuate spiking activity and overall circuit activity). In each GM neuron (N = 10), we used two-electrode current clamp at the soma to measure voltage responses arising from glutamate receptor activation at the soma, primary neurite, and more distal sites (*Figure 4*; 7–20 photo-uncaging sites per neuron with 1 ms, 30 mW UV pulses and a bath concentration of 250 µM MNI-glutamate). Experiments probing the spatial resolution of our photo-uncaging system demonstrated that the effective photo-uncaging radius was approximately 15 µm, a resolution high enough to target individual neurites (*Figure 4—figure supplement 1A–D*).

Focal glutamate photo-uncaging evoked hyperpolarizing voltage deflections across the neuronal structure (*Figure 4*). These inhibitory potentials are consistent with ionotropic glutamate-gated inhibitory currents described in previous work (*Marder and Paupardin-Tritsch, 1978*; *Eisen and Marder, 1982*; *Marder and Eisen, 1984*; *Cleland and Selverston, 1995*). Although voltage responses were uniformly inhibitory across the neuronal structure, their amplitudes varied (*Figure 4A–C*). *Figure 4A–C* show dye-fills of three individual GM neurons with photo-uncaging sites indicated with colored circles (left) and maximal responses to focal glutamate photo-uncaging that vary in amplitude across these sites (right). These focal glutamate responses were relatively stable, desensitizing only when the inter-pulse period (IPPs) was less than 10 s (*Figure 4—figure supplement 1E–G*).

Across GM neurons (N = 10), a non-zero response (measured at the soma) to glutamate applied to the soma was observed in only one GM neuron. Response amplitudes varied across positions with a mean maximum amplitude of 2.15 ± 1 mV. The mean coefficient of variation of non-zero responses, across preparations, was 0.4 ± 0.1 (N = 10), suggesting that, on average, non-zero responses varied by approximately 40% of the mean response amplitude within each neuron.

The electrotonic decrement of a voltage signal, from photo-stimulation site to recording site, is dependent on a number of factors: the diameter of the neurite through which it propagates (which influences the axial resistance to current flow), the membrane resistance, the extent of branching and associated loss of conductance at branch points, and the absolute distance the signal travels

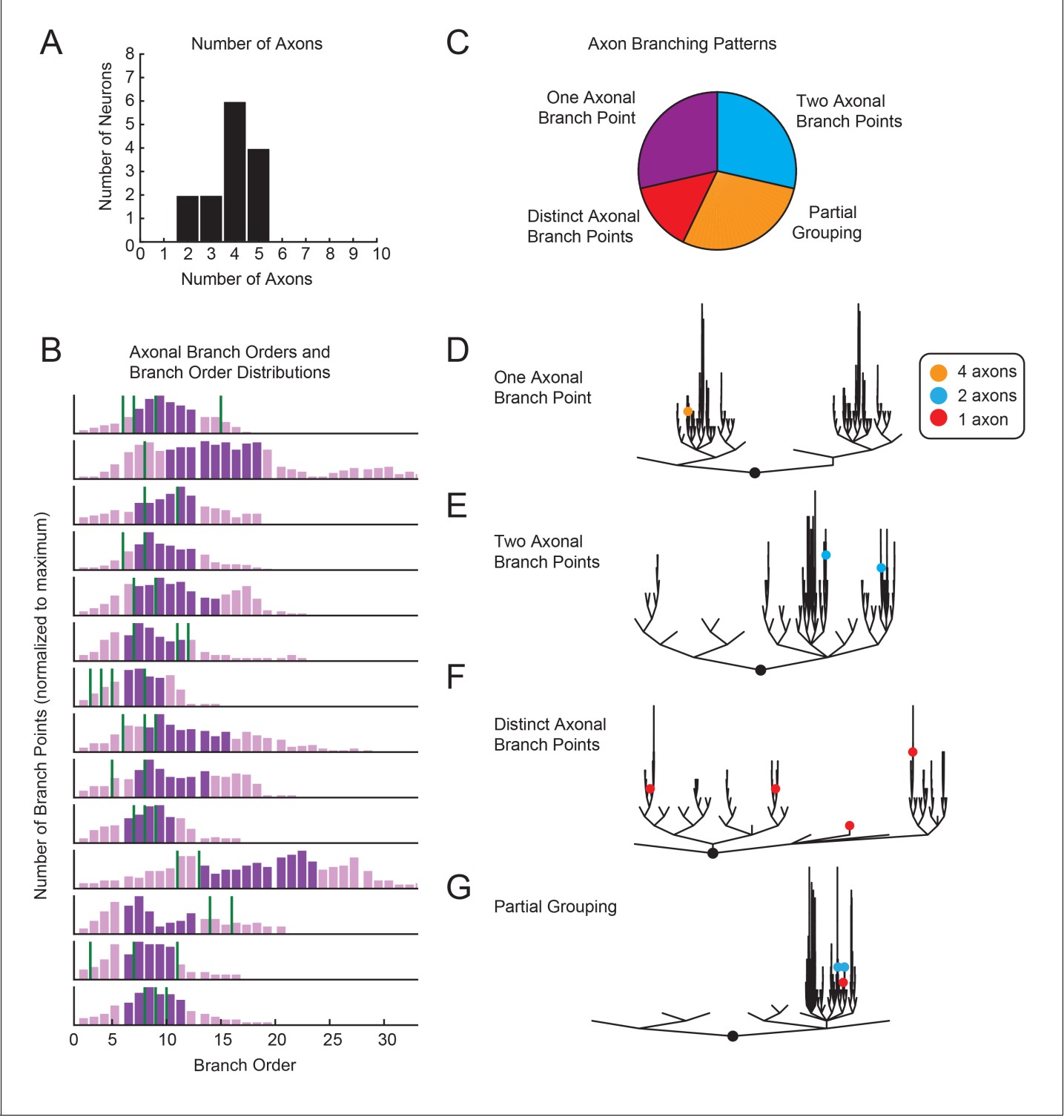

**Figure 3.** Variable axon location and branching patterns in GM neurons. (**A**) Histogram shows the number of distinguishable axonal projections for 14 GM neurons. (**B**) Histograms show the distribution of branch orders across 14 GM neurons. Axonal branch point orders are indicated with green lines. Dark purple shading indicates the range between the 25th and 75th percentiles. For easy comparison across neurons, histograms were plotted with 100 bins and y axes were normalized to the maximum number of branch points given the 100 bins. (**C**) Pie chart summarizes axonal branching patterns across 14 GM neurons. (**D–G**) Examples of the different axonal branching pattern possibilities in **C** are illustrated with dendrograms of four GM neurons from the tested population. Axonal branch points are indicated with colored circles indicating the number of axons projected from that branch point: orange = 4 axons, blue = 2 axons, red = 1 axon.

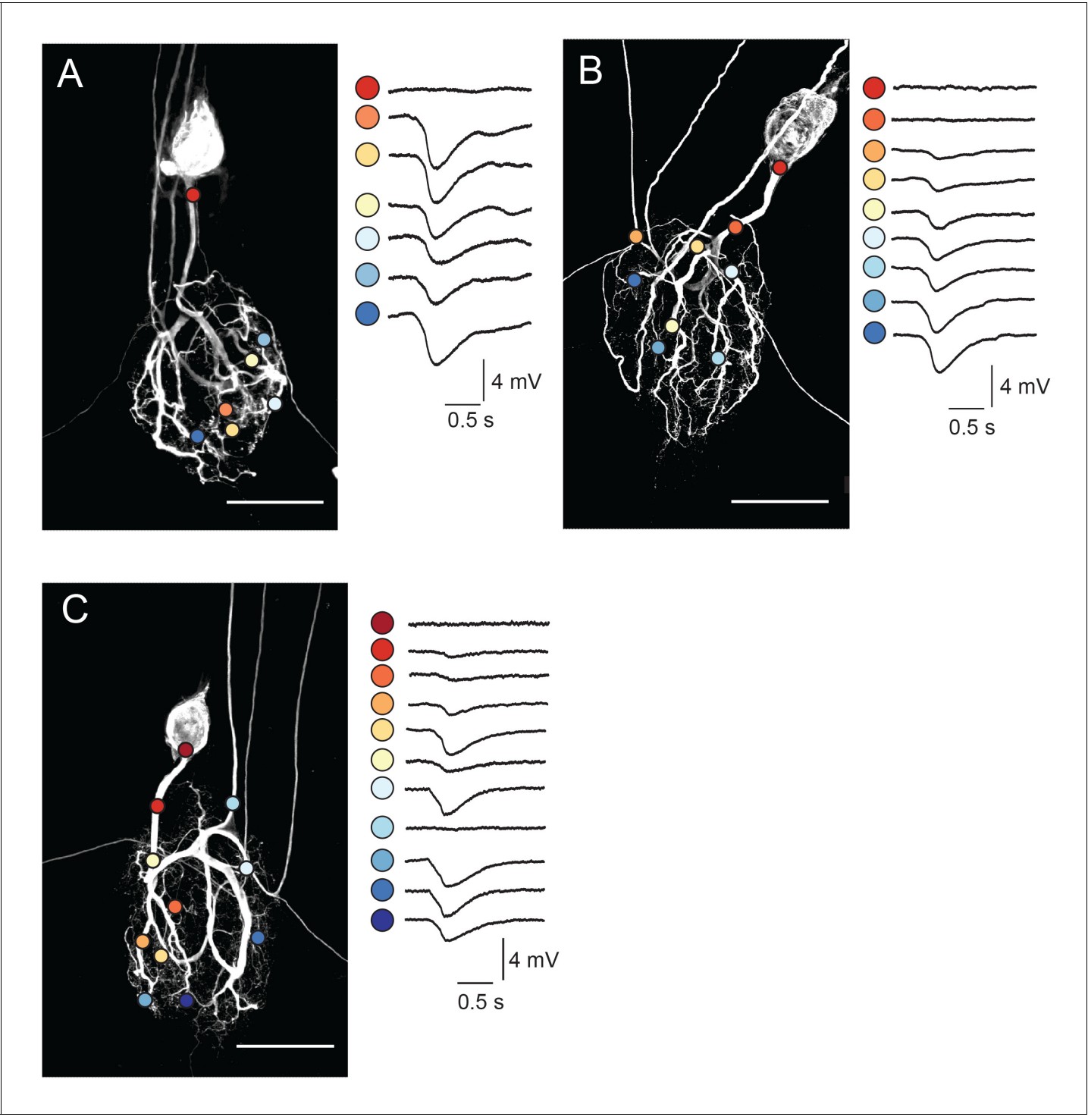

**Figure 4.** Focal glutamate responses across GM neuronal structures. (A–C) show maximum z-projections of confocal stacks of neuronal dye-fills with photo-uncaging sites indicated with colored circles. In each case, raw traces are shown on the right for maximal inhibitory glutamate responses evoked with 1 ms pulses at 30 mW UV laser intensity while. holding the membrane potential at −40 mV using two-electrode current clamp at the somatic recording site. Note the variability in response amplitude across different photo-uncaging sites within each neuron. Image scale bars are 150 µm.

The following figure supplement is available for figure 4:

**Figure supplement 1.** Focal glutamate photo-uncaging resolution and response desensitization.

(**Rall, 1959**; **Goldstein and Rall, 1974**). To quantify the dependence of response amplitude on each neuron's cable properties, we generated 'lolliplots' of response amplitude, measured at the soma, to each photo-stimulated site, as a function of their distance from the somatic recording site, branch order, and neurite size (measured in terms of neurite diameter in the x-y plane; **Figure 5**). The lolliplots show that the greatest response amplitudes can occur from photo-stimulation at sites with any distance from the soma, of any branch order, and of any size. Linear regression analyses of response amplitudes as a function of each of these geometric parameters showed no significant linear relationships, with insignificant p-values>0.1 (**Figure 5—figure supplement 1**, **Table 1**). Taken together, these data suggest that there is no apparent dependence of somatic response amplitude on where the responses were evoked.

In each neuron, heterogeneous response amplitudes are likely a consequence of variable receptor densities and cable properties. Thus, simply measuring the maximal response amplitudes arising from photo-stimulation at each of these sites and regressing them against their varying cable properties does not negate the possibility that the measured amplitudes are a consequence of both of these factors.

## Probing electrotonic structure with reversal potential measurements

A neuron's distributed cable properties, or electrotonic structure, can be quantitatively assessed by measuring the reversal potentials (more specifically, the *apparent* $E_{rev}$s measured at the soma) of local glutamate responses evoked at sites varying in distance from the somatic recording site (**Calvin, 1969**; **Carnevale and Johnston, 1982**). This approach distinguishes the degree to which the passive cable properties cause electrotonic decrement of voltage signals in their path of propagation to the somatic recording site, independent of the maximal conductance or receptor density at the photo-stimulation site. To demonstrate this logic, we built a library of passive cable models in the NEURON simulation platform (**Hines and Carnevale, 1997**; see Materials and methods) and simulated activation of local inhibitory chloride currents (actual $E_{rev}$ = −70 mV) at varying distances from the recording site. Apparent $E_{rev}$s were measured by manipulating the membrane potential ($V_m$) with current injections between −8 and +2 nA at the recording site. First, we demonstrate that the apparent $E_{rev}$ of an inhibitory event evoked 200 μm away from the recording site (as shown in **Figure 6A**) is independent of its maximal conductance (or receptor density; **Figure 6B–C**). The apparent $E_{rev}$s were measured with four maximal conductance ($g_{max}$) values: 1, 5, 10, and 50 nS. Although the response amplitudes measured at the recording site vary with $g_{max}$, all voltage responses flip their sign at −77 mV (**Figure 6B**). This is shown graphically in **Figure 6Ci**, where response amplitude (deltaV) is plotted as a function of $V_m$ (as measured at the recording site at 0 μm). For each $g_{max}$ value, the apparent $E_{rev}$ was calculated as the x-intercept of the linear regression of the deltaV versus $V_m$ curve (R > 0.9 and p<0.01 in each case). The inset in **Figure 6Ci** and **Figure 6Cii** clearly show that the x-intercepts are the same for all four $g_{max}$ values. The apparent $E_{rev}$ is independent of $g_{max}$ magnitude, regardless of the activation site's distance from the recording site (**Figure 6—figure supplement 1**).

Second, we show that the apparent $E_{rev}$ for a locally activated inhibitory current changes as a function of distance from the recording site (**Figure 6D–F**). The apparent $E_{rev}$s were measured for inhibitory currents of the same $g_{max}$ (5 nS) but with increasing distance from the recording site: 0 to 1000 μm, at increments of 200 μm (**Figure 6D**). It is evident from the traces in **Figure 6E** that the apparent $E_{rev}$, or membrane potential at which the responses change sign, occurs at increasingly hyperpolarized membrane potentials as a function of increasing distance from the recording site. This hyperpolarizing shift in apparent $E_{rev}$ is illustrated in **Figure 6Fi**, where the x-intercepts for the deltaV versus membrane potential curves (as measured at the recording site at 0 μm) shift leftward with increasing activation site distance (shown clearly in the inset, a magnification of the x-intercepts). The apparent $E_{rev}$ shifts from an accurate measure of the actual $E_{rev}$ of −70 mV when activated at the recording site (0 μm) to −110 mV when activated 1000 μm away (**Figure 6Fii**). Thus, from considering one cable model with one set of passive properties, we can deduce that the apparent $E_{rev}$ of a locally-activated inhibitory current is independent of its maximal conductance (or receptor density) and is dependent on the site's distance from the recording site.

**Figure 7** demonstrates that the dependence of the apparent $E_{rev}$ on distance is contingent upon the effective electrotonic structure of the voltage signal's path of propagation, from activation site to recording site. With a similar set-up to the simulation shown in **Figure 6C–F**, apparent $E_{rev}$s were

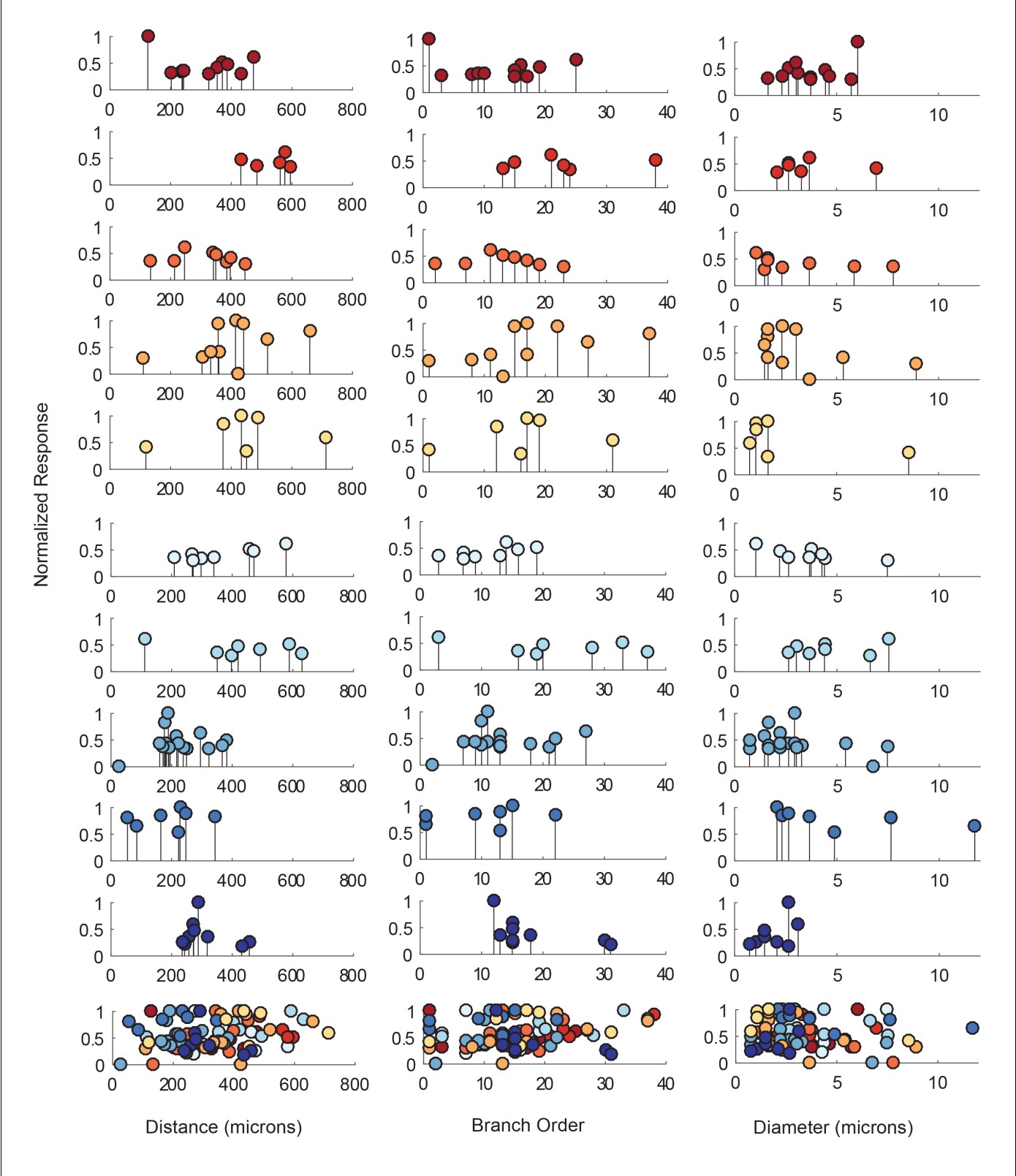

**Figure 5.** Response amplitudes as a function of various cable properties. Each lolliplot shows the normalized maximal response amplitudes for photo-uncaging sites that vary in distance from the soma, branch order, and neurite diameter. Each color is indicative of focal responses from a single neuron (N = 10). Focal glutamate responses were evoked at a depolarized membrane potential (−40 or −50, constant within each neuron), in two-electrode current clamp at the soma. Scatter plots at the bottom show these same data from all 10 neurons on the same axes. To allow comparison across

*Figure 5 continued on next page*

*Figure 5 continued*

neurons, voltage responses. were normalized to the maximum voltage response within each preparation. Distance and branch order measurements were generated from skeletal reconstructions. Diameter measurements were measured manually from neuronal dye-fill confocal stacks. Results from linear regression analyses of the above data are shown in the *Figure 5—figure supplement 1* and in *Table 1*.

The following figure supplement is available for figure 5:

**Figure supplement 1.** Response amplitudes as a function of various cable properties with raw response amplitudes and linear fits.

measured for responses arising from activation of the inhibitory current at sites with increasing distance from the recording site (*Figure 7A*). We conducted this simulation in a library of cable models with uniform diameters (5 µm) and lengths (1000 µm), but varying passive properties. Specifically, this set of 20 cable models varied in their combinations of passive leak conductances ($g_{pas}$): 5, 10, 20, or 50 nS/cm$^2$ and axial resistances ($R_a$): 1, 5, 10, 30, or 50 (Ω·cm). Thus, each cable model is distinguished by its electrotonic length constant ($\lambda$), determined by the expression: $\lambda = \sqrt{\frac{rR_m}{2R_a}}$ where $R_m = \frac{1}{g_{pas}}$. $\lambda$, in µm or mm, is equivalent to the distance at which a propagating voltage signal decrements to 37% of the maximal voltage signal (as would be measured at the site of activation). *Figure 7B* shows inhibitory voltage events evoked at increasing distances from the recording site (at 0 µm) in three cable models with different $\lambda$ values (200, 460, and 800 µm). With distance, the apparent $E_{rev}$ undergoes a hyperpolarizing shift (as was the case *Figure 6E*). However, the rate at which this apparent $E_{rev}$ hyperpolarizes, as a function of distance, is dependent on the electrotonic length constant of the cable (*Figure 7C*). When $\lambda$ = 800 µm, all apparent $E_{rev}$s are −70 mV, regardless of activation site distance. When $\lambda$ = 460 µm, apparent $E_{rev}$s shift from −70 mV at the recording site (0

**Table 1.** Linear regression analysis for response amplitudes as a function of distance, branch order, and neurite diameter. Each row corresponds to a different GM neuron, with same color scheme, as shown in *Figure 5* and *Figure 5—figure supplement 1*. Note insignificant p values suggesting no dependence of the response amplitude on these cable properties. n corresponds to the number of photo-uncaging sites in each GM neuron.

| Neuron | Distance | | | | Branch order | | | | Diameter | | | | n |
|---|---|---|---|---|---|---|---|---|---|---|---|---|---|
| | MSE | R | p | slope (mV/um) | MSE | R | p | slope (mV/order) | MSE | R | p | slope (mV/um) | |
| ● | 0.11 | −0.27 | 0.418 | −0.0009 | 0.12 | −0.20 | 0.55 | −0.0104 | 0.10 | 0.39 | 0.24 | 0.1028 | 11 |
| ● | 0.02 | 0.28 | 0.592 | 0.0003 | 0.02 | 0.26 | 0.62 | 0.0052 | 0.03 | 0.01 | 0.99 | 0.0007 | 7 |
| ● | 0.03 | −0.19 | 0.656 | −0.0003 | 0.03 | −0.21 | 0.62 | −0.0057 | 0.02 | −0.44 | 0.28 | −0.0331 | 9 |
| ● | 0.59 | 0.42 | 0.229 | 0.0026 | 0.51 | 0.54 | 0.11 | 0.0472 | 0.57 | −0.45 | 0.19 | −0.1713 | 11 |
| ● | 0.14 | 0.22 | 0.679 | 0.0005 | 0.14 | 0.23 | 0.66 | 0.0100 | 0.12 | −0.48 | 0.33 | −0.0680 | 7 |
| ○ | 0.01 | 0.90 | 0.002 | 0.0013 | 0.02 | 0.68 | 0.07 | 0.0232 | 0.01 | −0.75 | 0.03 | −0.0728 | 9 |
| ○ | 0.02 | -−0.49 | 0.269 | −0.0005 | 0.03 | −0.44 | 0.33 | −0.0073 | 0.03 | 0.32 | 0.48 | 0.0338 | 8 |
| ● | 0.11 | 0.20 | 0.443 | 0.0008 | 0.11 | 0.24 | 0.35 | 0.0142 | 0.10 | −0.37 | 0.14 | −0.0682 | 18 |
| ● | 0.02 | 0.26 | 0.579 | 0.0004 | 0.02 | 0.32 | 0.49 | 0.0060 | 0.01 | −0.60 | 0.16 | −0.0244 | 8 |
| ● | 0.32 | −0.27 | 0.477 | −0.0021 | 0.25 | −0.53 | 0.15 | −0.0455 | 0.25 | 0.51 | 0.16 | 0.3891 | 11 |
| Mean | 0.14 | 0.11 | 0.434 | 0.0002 | 0.12 | 0.09 | 0.39 | 0.0037 | 0.12 | −0.19 | 0.30 | 0.0089 | |
| SD | 0.18 | 0.41 | 0.214 | 0.0013 | 0.15 | 0.41 | 0.23 | 0.0242 | 0.17 | 0.45 | 0.27 | 0.1520 | |

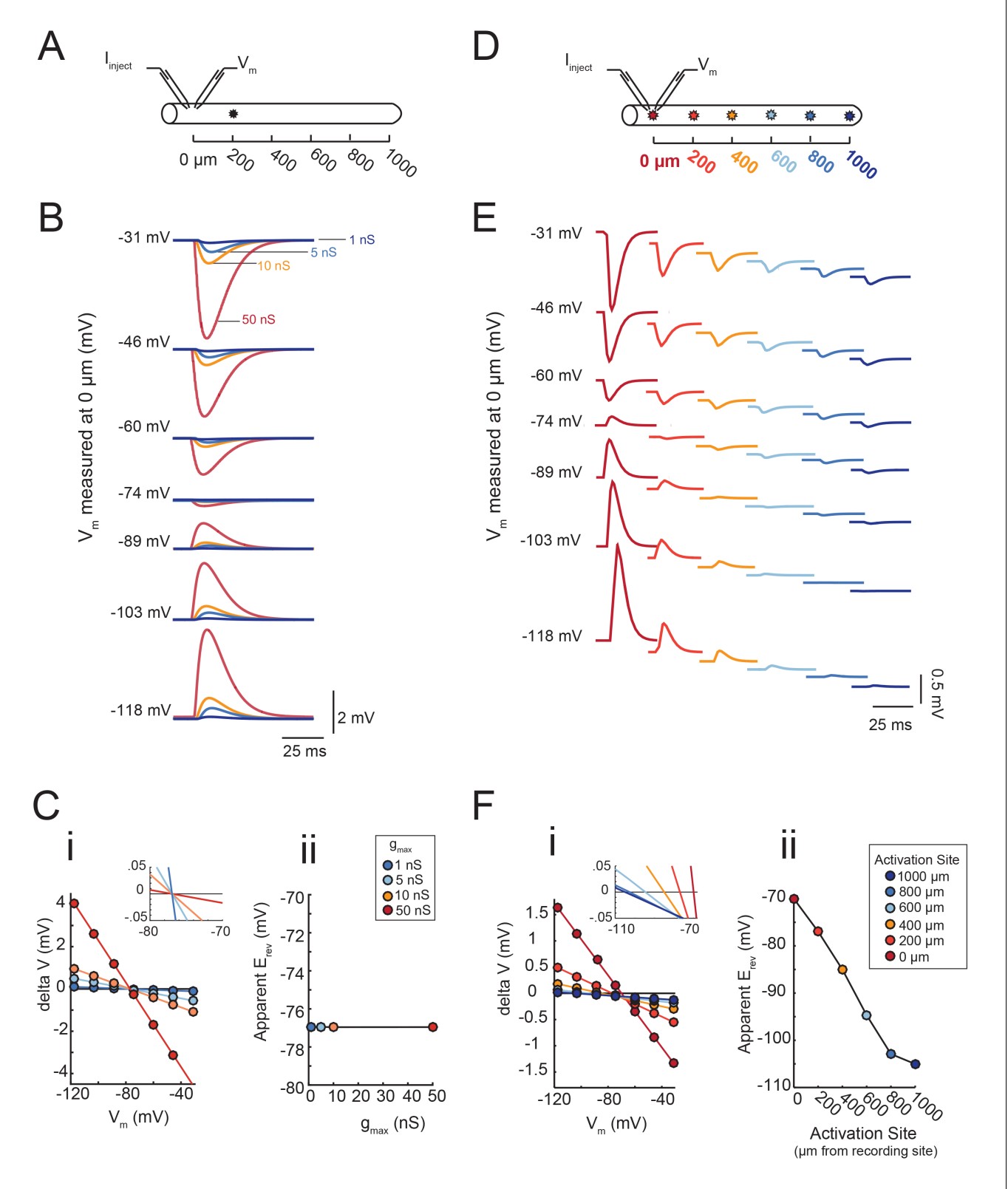

**Figure 6.** Passive cable simulations show that apparent $E_{rev}$ measurements are independent of maximal conductance ($g_{max}$) and dependent on the distance between activation and recording sites. (A) An inhibitory current (actual $E_{rev}$ = −70 mV, $\tau$ = 3 ms) was activated 200 μm from the recording site (at 0 μm). (B) Voltage events as measured at 0 μm. The membrane potential at the recording site was manipulated with current injections between −8 and + 2 nA. Colors correspond to responses evoked at 200 μm with different $g_{max}$ values (as indicated). (C) (i) Response amplitude (deltaV) plotted as a

*Figure 6 continued on next page*

*Figure 6 continued*

function of membrane potential ($V_m$) as measured at 0 µm. Apparent $E_{rev}$s were identified for each $g_{max}$ by calculating the x- intercept of linear fits to these curves (R > 0.9 and p<0.01 in all cases). The inset shows a magnification along the x-axis and illustrates that all curves share the same x-intercept, regardless of. $g_{max}$ value. (ii) Apparent $E_{rev}$ plotted as a function of $g_{max}$ values. (D) To determine the dependence of apparent $E_{rev}$ as a function of activation site distance from recording site, inhibitory currents ($E_{rev}$ = −70 mV, τ = 3 ms, $g_{max}$ = 5 nS) were evoked at 0, 200, 400, 600, 800,. and 1000 µm from the recording site. (E) Voltage events as measured at 0 µm. The membrane. potential at the recording site was manipulated with current injections between −8 and + 2 nA. Colors correspond to responses evoked at different activation sites (as indicated in D). (F) (i) Response amplitude (deltaV) plotted as a function of membrane potential ($V_m$) as measured at 0 µm. Apparent $E_{rev}$s were identified for each activation site distance by calculating the x-intercept of linear fits to these curves (R > 0.9 and p<0.01 in all cases). The inset shows a magnification along the x-axis and illustrates the hyperpolarizing shift in apparent $E_{rev}$ as activation site distance increases. This is explicitly plotted in (ii). The cable model used to generate all of these data had a passive conductance of 20 nS·cm$^{-2}$ and axial resistance of 30 Ω·cm (see Materials and methods).

The following figure supplement is available for figure 6:

**Figure supplement 1.** The apparent $E_{rev}$ remains independent of $g_{max}$, regardless of the activation site's distance from the recording site.

µm) to −88 mV at 1000 µm. When λ = 200 µm, the apparent $E_{rev}$ shifts from −70 mV at 0 µm to well below −100 mV when the activation site is 1000 µm away. It is evident that the apparent $E_{rev}$ undergoes a greater hyperpolarizing shift with distance with decreasing λ values. *Figure 7D* shows the dependence of apparent $E_{rev}$ as a function of activation site location for all 20 cable models with λ values ranging between 200–5000 µm. Considering a subset of these cable models, with λ values between 300–1700 µm, the apparent $E_{rev}$s measured at different activation site distances diverge as λ decreases to 300 µm and converge as λ increases beyond 1000 µm (*Figure 7E*).

Taken together, these proof-of-principle simulations show the unmistakable relationship between electrotonic structure and apparent $E_{rev}$s measured for activation sites varying in their distance from the recording site. Invariant apparent $E_{rev}$s suggest a relatively high electrotonic length constant, whereas heterogeneous $E_{rev}$s suggest a lower electrotonic length constant. As is shown in *Figure 6*, this approach to characterizing the electrotonic structure, or passive cable properties, of a neuron is independent of differential $g_{max}$ values, or receptor densities, across the neuronal structure.

In this simulation paradigm, current was injected at the recording site and flowed from the recording site to the stimulation site, changing the membrane potential at the distal site. The difference between the apparent $E_{rev}$ and the actual $E_{rev}$ is indicative of the ease of current flow in this direction and ability to manipulate the membrane potential at this distal site. Thus, there will always be a discrepancy between the voltage at the soma and the voltage at the photo-uncaging site and this discrepancy will depend on the effective electrotonic length constant of the neurite path. Therefore, this assay tests the passive cable properties of the path of propagation as is most relevant to current flow from recording site to stimulation site. Even so, the observations of sizeable voltage events at the recording site and reasonable reversal potentials are suggestive of a level of electrotonic compactness that is relevant to voltage signal propagation in either direction.

## Distributed reversal potentials in GM neurons are nearly invariant

Using two-electrode current clamp at the soma, we measured apparent $E_{rev}$s of local inhibitory responses evoked by focal photo-uncaging of glutamate at positions varying in distance from the somatic recording site. In these experiments, current was injected at the somatic recording site and flowed centrifugally from the recording site to the photo-uncaging site, changing the membrane potential at this distal site. *Figure 8* illustrates the apparent $E_{rev}$s of local inhibitory responses evoked at 7–15 sites, varying in their cable properties, across the same GM neuronal structure described in *Figure 4C* (a second example is shown in the *Figure 8—figure supplement 1*). For each photo-uncaging site, response amplitudes measured at the soma were plotted as a function of somatic membrane potential (*Figure 8B,C*; *Figure 8—figure supplement 1B and C*). These data were fit with linear regression analyses (R > 0.9 in all cases) and the reversal potentials were determined by calculating the x-intercepts of the linear fits. The linear fits for each position show little variation in the reversal potential across all positions within each preparation, with a within-neuron coefficient of variation of 0.04 ± 0.01 (mean ± SD; *Figure 8D*; *Figure 8—figure supplement 1D*). Although apparent $E_{rev}$s were nearly invariant within each neuron, mean reversal potentials across

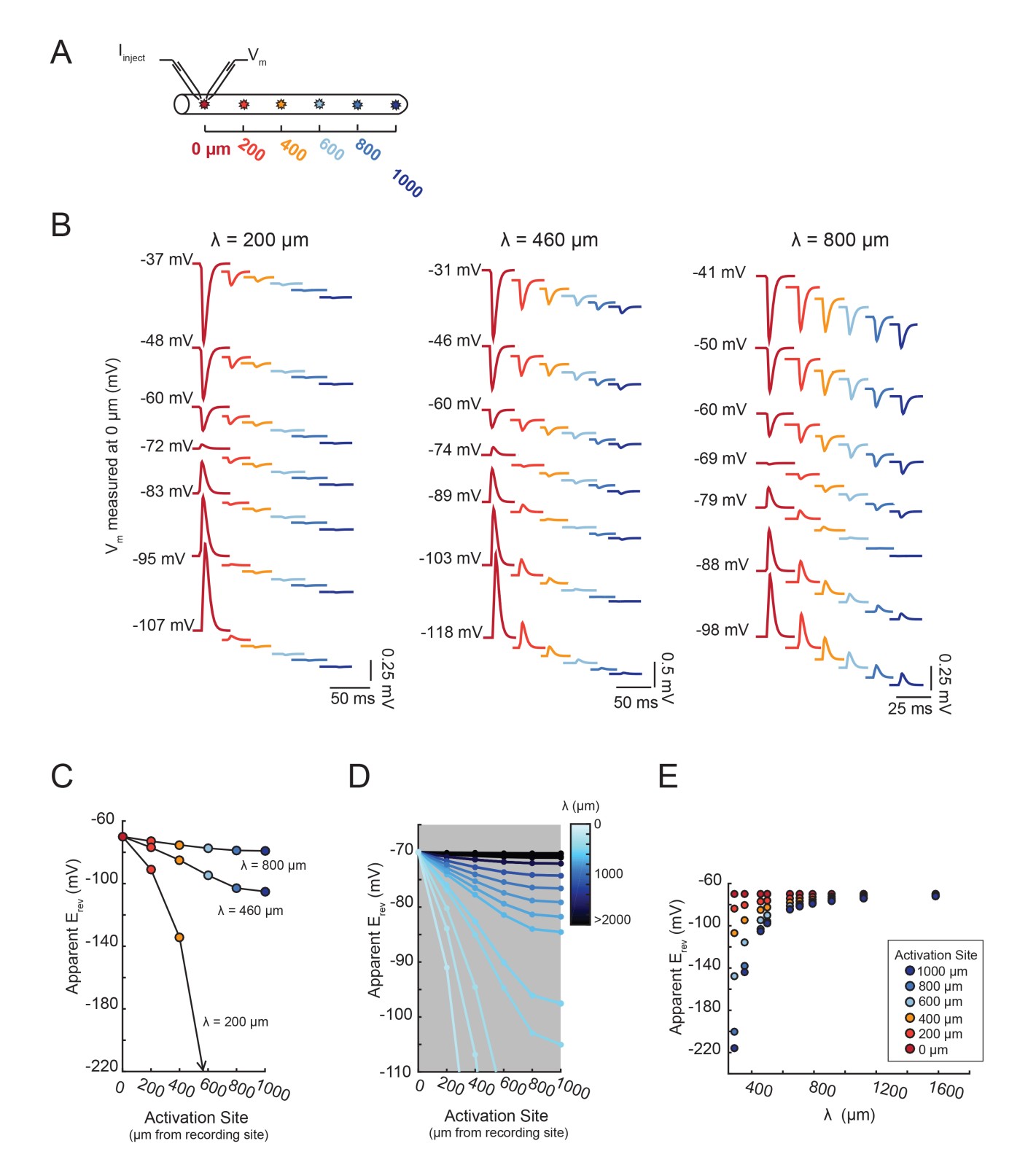

**Figure 7.** The shift in apparent E_rev with activation site distance is contingent upon the electrotonic length constant, λ. (**A**) For each of the 20 cable models, with varying passive properties and, consequently, λ values (see Materials and methods), an inhibitory current (actual E_rev = −70 mV) was evoked at varying distances from the recording site 0 μm. (**B**) Voltage events as measured at 0 μm for cables with λ = 200, 460, and 800 μm. The membrane potential at the recording site (V_m) was manipulated with current injections between −8 and + 2 nA. Colors correspond to responses
*Figure 7 continued on next page*

*Figure 7 continued*

evoked at different activation sites (as indicated in **A** and *Figure 6D*). With increasing activation site distance, the voltage deflections flip sign at increasingly hyperpolarized $V_m$s at the recording site. This hyperpolarizing shift is more drastic in the cable with the lowest $\lambda$. (**C**) Plot of the apparent $E_{rev}$ (measured at 0 µm) as a function of activation site distance, in the three cables shown in **B**. The colors are indicative of the activation site distance, as in the traces in **B**. (**D**) Plot of apparent $E_{rev}$ (measured at 0 µm) as a function of activation site distance for all 20 cable models. The blue colorbar indicates relative $\lambda$ values, ranging between 200 µm and 5 mm. (**E**) Plot of apparent $E_{rev}$ (measured at 0 µm) as a function of $\lambda$, as measured in cables with $\lambda$s between 300 and 1600 µm. As $\lambda$ increases, apparent $E_{rev}$s of responses evoked at different activation site distances converge toward the actual $E_{rev}$.

neurons did vary, with a pooled mean of –78.6 ± 7 mV (*Table 2*). This could be attributed to real differences in the ionic current, which could be carried by potassium, chloride, or a combination of the two ions (*Marder and Paupardin-Tritsch, 1978*; *Eisen and Marder, 1982*). A consequence of variable response amplitudes across positions yielded variable linear fit slopes within neurons (as shown in *Figure 8D* and *Figure 8—figure supplement 1D*). This is likely a reflection of variable receptor densities or maximal conductances at the different photo-uncaging sites (consistent with the simulations shown in *Figure 6A–C*). There was no significant linear relationship between input resistances measured at the soma (*Table 2*) and the mean within-neuron reversal potentials (p=0.2; data not shown graphically, but available in *Table 2*).

Lolliplots for each preparation, showing apparent $E_{rev}$s for each photo-uncaging site as a function of their distance from the somatic recording site, diameter, and branch order (*Figure 9*), confirm no dependence of the apparent $E_{rev}$ on these cable properties. Linear regression analyses revealed near-zero slopes for all 10 neurons (*Figure 9—figure supplement 1*; *Table 2*). This suggests that, even though these sites vary in their absolute distance, diameter, and branch order, they do not vary substantially in their electrotonic distance from the somatic recording site. The minimal hyperpolarizing shift for apparent $E_{rev}$s measured for activation sites between 100 and 800 µm from the recording site is consistent with a $\lambda > 1.5$ mm (referencing *Figure 7D and E*). Taken together, these results demonstrate that GM neurons are surprisingly electrotonically compact, despite their expansive structures and morphological complexity.

## Discussion

Neuronal circuits function reliably despite remarkable animal-to-animal variability in the synaptic and intrinsic conductances of their constituent neurons (*Goaillard et al., 2009*; *Norris et al., 2011*; *Roffman et al., 2012* ; *Sakurai et al., 2014*); see *Calabrese et al. (2011)* and *Marder et al. (2015)* for reviews). Given that neuronal physiology also depends on the passive cable properties arising from geometry, we examined the physiological consequences of animal-to-animal variability in neuronal morphology. Despite their expansive and complex morphologies, GM neurons have electrotonically compact structures. This effectively compensates for morphological variability and contributes to consistent neuronal and circuit-level function across animals.

### Complex yet compact

Here, we present the ostensible conundrum wherein an identifiable neuron type, despite its complex, highly-branched, neurite tree, is surprisingly electrotonically compact. This result differs from studies in a variety of neuron types that attribute specific physiological computations and plasticity rules to compartmentalized electrotonic structures. Early work in insect identified neurons with stereotyped dendritic branching patterns showed that electrotonically distinct dendritic subtrees result in the weighted integration of sensory inputs (*Murphey et al., 1984*; *Bacon and Murphey, 1984*; *Miller and Jacobs, 1984*; *Jacobs and Miller, 1985*). Studies in hippocampal pyramidal neurons (*Spruston and Johnston, 1992*; *Carnevale et al., 1997*; *Mainen and Sejnowski, 1996*; *Jaffe and Carnevale, 1999*), have attributed Hebbian plasticity in part to the passive normalization of postsynaptic potentials arising from the electrotonically distant apical and basal dendritic tufts. Work in medium spiny neurons (*MacAskill et al., 2012*) and thalamocortical neurons (*Connelly et al., 2016*) have demonstrated that the activation pattern of spatially-distributed, electrotonically distant, synaptic inputs produces different neuronal and circuit-level computations. The present work differs from

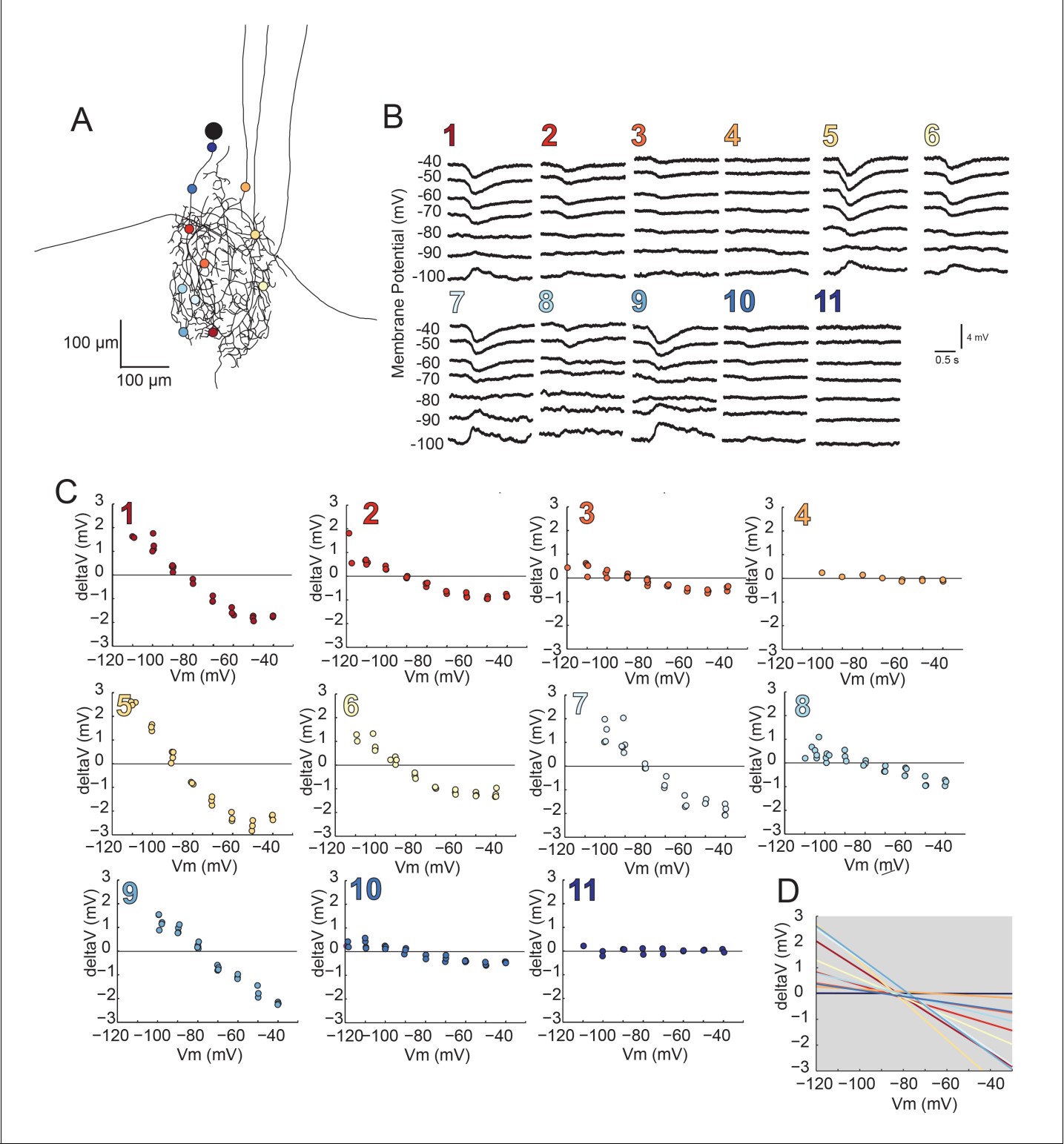

**Figure 8.** Reversal potentials are nearly invariant across individual neuronal structure. (A) Photo-uncaging sites are indicated as unique colors on the skeletal reconstruction of one GM neuron. (B) Raw voltage traces show focal glutamate responses as measured at the soma in two-electrode current clamp when evoked at positions indicated in A. (C) Plots of response peak amplitude (deltaV) as a function of membrane potential at soma (Vm) for each photo-uncaging site. (D) Linear functions for each position generated by linear regression analyses of data shown in C. Note that the x-intercepts (apparent $E_{rev}$s) are highly invariant across photo-uncaging sites. The mean apparent $E_{rev}$ + SD = −84.6 + 4.3 mV and coefficient of variation (CV) is 0.05 in this neuron.

*Figure 8 continued on next page*

*Figure 8 continued*

The following figure supplement is available for figure 8:

**Figure supplement 1.** Reversal Potentials are nearly invariant across individual neuronal structure, a second example.

these other studies of neuronal electrotonus and enriches our framework for understanding how morphology maps (or does not map) to physiological function. Otherwise stated: structural complexity does not necessarily yield compartmentalized computations.

## Physiological implications

In GM neurons, synaptic voltage events may propagate tortuous neurite paths that extend beyond half a millimeter in length (*Figure 2*). Yet, current can be injected at the somatic recording site and effectively alter the membrane potential at such distal and distant sites allowing for reasonable apparent $E_{rev}$s (approximately −80 mV). Likewise, the amplitude of the voltage response at the photo-uncaging site must be larger than what we observe at the soma. Because distally evoked events, initiated as far as 1 mm away, can be observed at the somatic recording site, the electrotonic decrement of these distal events must be small enough so that apparent reversal potentials can be recorded. Lastly, the invariance in apparent $E_{rev}$s across sites (a measure independent of receptor density) suggests relatively little variance in the electrotonic decrement of signals coming from disparate sites across the neurite tree. In this sense, these neurons function almost like a single compartment, despite their complex structures.

**Table 2.** Linear regression analysis for reversal potentials as a function of distance, branch order, and neurite diameter. Each row corresponds to a different GM neuron, with same color scheme, as shown in *Figure 8* and *Figure 8—figure supplement 1*. Note slopes of nearly zero across all 10 neurons, suggesting invariant reversal potentials across the neuronal structure. n corresponds to the number of photo-uncaging sites in each GM neuron.

| Neuron | Reversal potentials | | | Distance | | | | Branch order | | | | Diameter | | | | n | $R_{input}$ (MΩ) |
| | Mean (mV) | SD | CV | MSE | R | p | slope (mV/um) | MSE | R | p | slope (mV/order) | MSE | R | p | slope (mV/um) | | |
|---|---|---|---|---|---|---|---|---|---|---|---|---|---|---|---|---|---|
| ● | −73.8 | 2.9 | −0.04 | 5.99 | 0.44 | 0.18 | 0.012 | 6.52 | 0.35 | 0.30 | 0.139 | 6.52 | 0.34 | 0.30 | 0.712 | 11 | 10 |
| ● | −79.2 | 1.2 | −0.02 | 1.21 | 0.00 | 0.99 | 0.000 | 1.16 | −0.21 | 0.68 | −0.029 | 0.54 | −0.75 | 0.09 | −0.513 | 6 | 15 |
| ● | −81.9 | 3.2 | −0.04 | 8.80 | 0.14 | 0.76 | 0.006 | 8.86 | 0.12 | 0.80 | 0.071 | 5.86 | −0.59 | 0.16 | −1.123 | 7 | 12 |
| ● | −83.6 | 4.3 | −0.05 | 13.13 | 0.45 | 0.23 | 0.013 | 13.35 | 0.43 | 0.24 | 0.175 | 14.39 | −0.35 | 0.35 | −0.613 | 9 | 12 |
| ● | −75.6 | 3.8 | −0.05 | 9.42 | 0.15 | 0.90 | 0.0010 | 9.61 | -0.05 | 0.97 | −0.058 | 0.34 | −0.98 | 0.12 | −10.756 | 3 | 11 |
| ● | −87.7 | 2.7 | −0.03 | 5.18 | 0.34 | 0.51 | 0.009 | 5.27 | 0.32 | 0.54 | 0.142 | 5.20 | 0.34 | 0.51 | 0.478 | 6 | 7 |
| ● | −64.8 | 3.6 | −0.06 | 8.59 | −0.50 | 0.26 | −0.011 | 7.53 | −0.58 | 0.17 | −0.185 | 11.38 | −0.04 | 0.93 | −0.084 | 7 | 5 |
| ● | −69.9 | 3.3 | −0.05 | 9.37 | −0.22 | 0.46 | −0.008 | 9.01 | −0.29 | 0.33 | −0.146 | 9.84 | 0.05 | 0.87 | 0.015 | 11 | 7 |
| ● | −84.6 | 3.0 | −0.04 | 7.88 | −0.02 | 0.97 | −0.001 | 7.80 | 0.10 | 00.83 | 0.039 | 7.88 | −0.0 | 0.96 | −0.020 | 7 | 10 |
| ● | −85.0 | 2.2 | −0.03 | 3.02 | −0.53 | 0.28 | −0.042 | 4.11 | 0.13 | 0.80 | 0.144 | 2.84 | 0.57 | 0.24 | 1.805 | 6 | 10 |
| Pooled Mean | −78.6 | | −0.04 | 7.26 | 0.03 | 0.55 | −0.001 | 7.32 | 0.03 | 0.57 | 0.029 | 6.48 | −0.14 | 0.45 | −1.010 | | 9.9 |
| Pooled SD | 7.4 | | | 3.47 | 0.35 | 0.32 | 0.016 | 3.34 | 0.32 | 0.29 | 0.128 | 4.57 | 0.51 | 0.34 | 3.518 | | 2.9 |

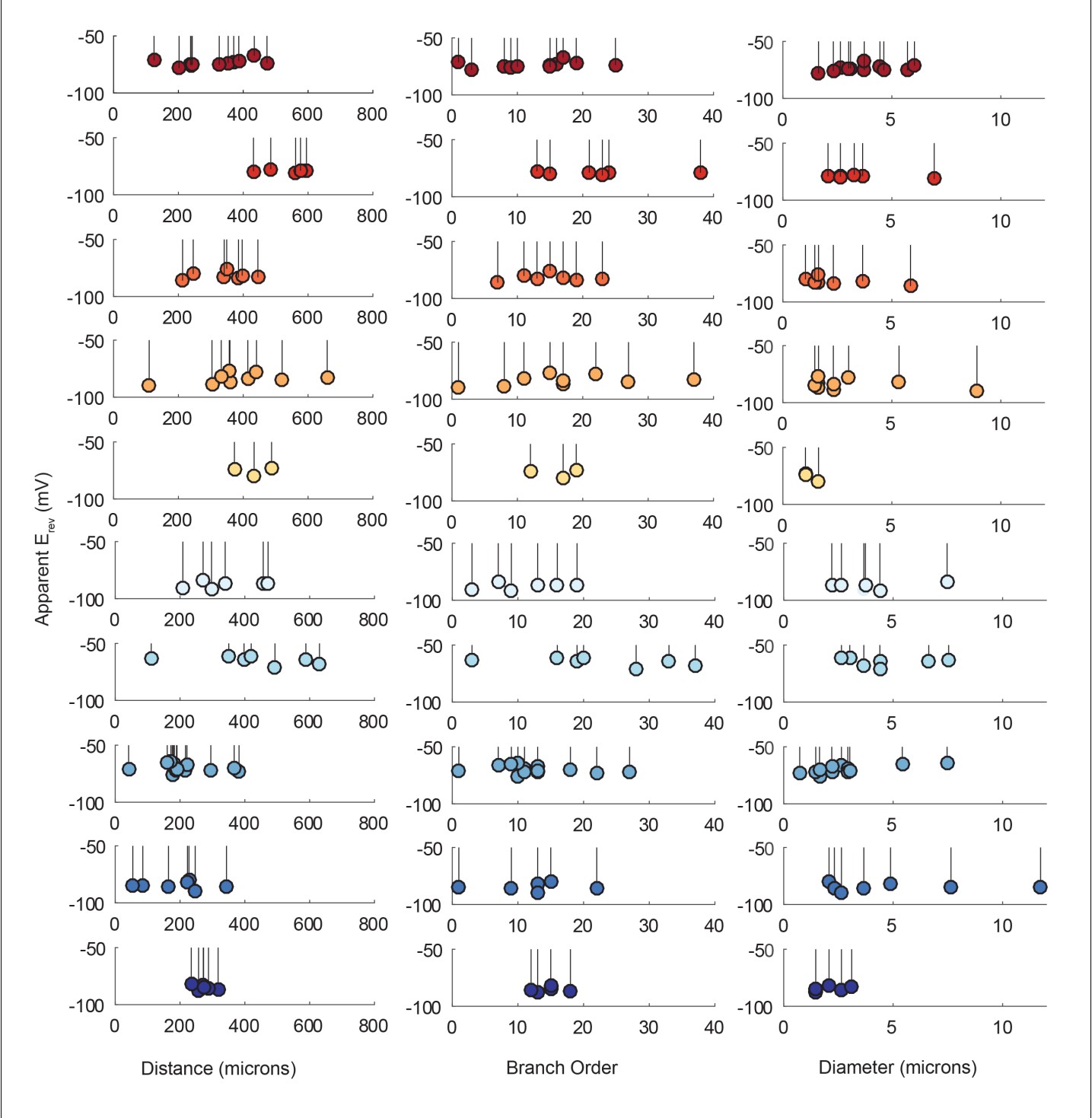

**Figure 9.** Reversal potentials as a function of various cable properties. Each lolliplot shows the glutamate response apparent reversal potentials ($E_{rev}$s) for photo-uncaging sites as function of their distance from the soma, branch order, and neurite diameter. Each color and lolliplot is indicative of the set of reversal potentials across sites within a single neuron (N = 10). Focal glutamate responses were evoked at varying membrane potentials (between −120 and −40 mV) with two-electrode current clamp at the soma. Apparent $E_{rev}$s for each photo-uncaging site were determined as in *Figure 6*. Distance and branch order measurements were generated from skeletal reconstructions. Diameter measurements were generated by manual measurement of neuronal dye-fill confocal stacks.

The following figure supplement is available for figure 9:

*Figure 9 continued on next page*

*Figure 9 continued*

**Figure supplement 1.** Reversal potentials as a function of various cable properties with linear fits.

The geometric and/or passive cable properties provide the most plausible explanation for this electrotonically compact structure. Recent work has quantified the fine anatomical properties of GM neurons and other STG neuron types, showing that the primary neurite can be as large as 15–20 μm in diameter and that the most distal neurite branches may taper to diameters between ≤1 and 10 μm (*Otopalik et al., 2017*). GM neurites may have relatively low axial resistances as a consequence of large neurite diameters. It is possible that voltage signal decrement may also be minimized by high membrane resistances across neurite branches. Given that these neurons are electrotonically compact, the input resistance as measured at the soma is likely a reflection of the resistance of the membrane surface area of much of the entire neuron. Thus, it is not surprising that input resistances measured at the soma are relatively low (mean of approximately 10 MΩ (*Table 2*), consistent with many years of recordings from STG neurons). If the neuron were less electrotonically compact, the input resistance measured at the soma would be higher, as the measurement would be restricted to the surface area of the local, somatic membrane.

Other neuron types compensate for passive attenuation of voltage responses with distance-dependent scaling of synaptic receptor density (*Andrasfalvy and Magee, 2001*; *Magee and Cook, 2000*; *Smith et al., 2003*). Our experiments showed heterogeneous maximal response amplitudes across the neuronal structure. Due to the uniformity of their apparent $E_{rev}$s, it is likely that these variable response amplitudes arise from local variations in receptor densities. Because the response amplitudes across sites vary in a manner that is independent of distance from the recording site (*Figure 5*; *Figure 5—figure supplement 1*; *Table 1*), it is unlikely that receptor densities are scaling with distance in a systematic way.

These experiments provide an 'upper bound' on the effective electrotonic length constant and, therefore, the compactness of GM neurons. These experiments were done in TTX, wherein TTX-sensitive, voltage-gated sodium channels are blocked and circuit activity is silenced. While the overall membrane conductance may be higher in the absence of TTX (although, no change in input resistance as measured at the soma was detected before and after TTX addition), we would not expect these TTX-sensitive currents to substantially alter inhibitory voltage signal propagation at the range of membrane potentials probed here (−120 to −40 mV). Furthermore, spike initiation zones, where TTX-sensitive channels are most likely to reside, are located in the periphery, where the axons exit the neuropil (*Raper, 1979*; *Miller, 1980*). Thus, it is unlikely that TTX-sensitive voltage-gated channels would shunt the current arising from these evoked events in the same way as has been seen in other systems (*Laurent, 1990*). In STG neurons, it is an intriguing possibility that separation of synaptic integration and slow waves from spike initiation zones may be a morphological strategy established to avoid shunting of synaptic currents. Future experiments in varying pharmacological and modulatory conditions could shed light on how different voltage-gated currents, modulatory currents, and ongoing synaptic input during rhythmic activity, may effectively compartmentalize these otherwise compact passive neuronal structures.

## Electrotonic structure in circuit context

GM electrotonic structure is best understood in light of STG circuit architecture. The identified neurons of the STG exhibit complex morphologies (*Wilensky et al., 2003*; *Baldwin and Graubard, 1995*; *Bucher et al., 2007*; *Goeritz et al., 2013*; *Otopalik et al., 2017*). Like the GM neuron, all STG neurons display large somata (50–150 μm in diameter) and primary neurites that ramify throughout the STG neuropil, wherein synaptic partners form numerous, sparse synapses (*King, 1976a*, *1976b*; *Baldwin and Graubard, 1995*). If each neuron type were highly electrotonically compartmentalized, yet variable across animals, wiring this circuit would be a puzzling developmental task. The fact that these neurons present electrotonically compact structures simplifies our understanding of the developmental wiring rules that may be required and how such a circuit can be relatively immune to structural differences.

The neurons of the STG rely on graded transmission and slow oscillations, rather than fast spikes, to maintain phase relationships at the circuit level (*Graubard, 1978*; *Raper, 1979*; *Graubard et al., 1980*, *1983*; *Anderson and Barker, 1981*; *Manor et al., 1997*, *1999*; *Bose et al., 2014*). Given the relatively slow temporal precision of this circuit, an electrotonically compact structure is sufficient for integrating activity of many, yet sparsely distributed, synaptic inputs from each presynaptic neuron (*King, 1976a*, *1976b*), independent of synaptic site locations. In this way, electrotonic compactness both masks the observed heterogeneity in glutamate sensitivity across the neuronal structure (*Figures 4* and *5*) and diminishes the consequences of presumed variability in synaptic site location arising from observed animal-to-animal variability in GM morphology (also see *Otopalik et al., 2017*).

In this scenario, presynaptic inputs may influence GM neuron activity with equivalent efficacy regardless of synaptic site location. This synaptic democracy (*Häusser, 2001*) is achieved by combining an electrotonically compact structure with graded transmission resilient to electrotonic decrement across sparsely distributed, synchronous presynaptic sites. This strategy is in stark contrast to the tight tuning of receptor or ion channel distributions employed by some neuron types. For example, CA1 pyramidal neurons compensate for passive attenuation of voltage responses with distance-dependent scaling of synaptic receptor density (*Andrasfalvy and Magee, 2001*; *Magee and Cook, 2000*; *Smith et al., 2003*). It is feasible that strategies for achieving synaptic democracy vary across circuit contexts. The input-output computations of pyramidal neurons are typically dependent on spikes and fast voltage transients, whereas the neurons of the stomatogastric ganglion rely more heavily on graded transmission and slow oscillations to serve their circuit-level function (*Graubard, 1978*; *Raper, 1979*; *Graubard et al., 1980*, *1983*; *Anderson and Barker, 1981*; *Manor et al., 1997*, *1999*; *Bose et al., 2014*). Electrotonic structure may reflect the temporal precision of the neuronal and circuit-level computations performed.

## Morphologies that are 'good enough' rather than optimal

Numerous works have argued that specific neuronal geometries are optimal for precise neuronal computations (*Mainen and Sejnowski, 1996*; *Stiefel and Sejnowski, 2007*; *Cuntz et al., 2010*). Experimentalists and theorists alike have suggested that neurons employ developmental growth rules that fine-tune neuronal geometry for optimal current transfer and wiring costs (*Chklovskii, 2000*, *2004*; *Chen et al., 2006*; *Wen and Chklovskii, 2008*; *Cuntz et al., 2007*, *2010*; *Kim et al., 2012*). Many of these works rely on studies in neuron types with both recognizable morphologies and known computations and/or plasticity rules. In this way, such rules hinge on a somewhat circular premise that specific neuronal functions arise from specialized geometries. As is evident in the present work, not all neuron types exhibit conserved morphologies across animals, yet show stereotyped physiological properties and circuit-level functions. We present a case in which the solution to the morphology-to-physiology transform is many-to-one.

## Meaning in morphology

Neuron types can have characteristic, recognizable morphologies. Many studies have explored stereotypy in macroscopic dendritic and axonal arborization patterns in a variety of systems, including cricket (*Miller and Jacobs, 1984*) and grasshopper (*Goodman, 1976*, *1978*) sensory interneurons, the insect (*Cuntz et al., 2008*) and mammalian (*Bloomfield and Miller, 1986*; *Hong et al., 2011*) retina, and somatosensory (*Wang et al., 2002*), motor (*Ghosh and Porter, 1988*), and visual (*Martin et al., 1983*; *Martin and Whitteridge, 1984a*, *1984b*) cortices. That said, there are remarkably few instances (*Cuntz et al., 2008*; *Wang et al., 2002*) in which multiple examples of relatively complete reconstructions have been published in enough detail to judge whether the ranges of neuronal morphological features shown here, pertinent to a neuron's cable properties, are typical or more pronounced than in other systems.

Here, we argue that the fine structural details of complex morphology may not matter for the neuronal and circuit-level function of an identified neuron type. In the STG, neurons rely predominantly on slow oscillations for circuit function, and electrotonically compact structures elegantly compensate for a high degree of animal-to-animal variability in morphology. The degree of animal-to-animal variability in neuronal morphology, and whether it is compensated for, may depend on the system and the precision of the neuronal and circuit-level computation(s) to be performed.

# Materials and methods

## Animals and dissections

Adult male Jonah Crabs (*Cancer borealis*) were purchased from Commercial Lobster (Boston, MA) and maintained in artificial seawater at 10–13°C on a 12 hr light/12 hr dark cycle without food. On average, animals were acclimated at this temperature for one week before use. Prior to dissection, animals were anesthetized for 30 min on ice. Dissections were performed as previously described (*Gutierrez and Grashow, 2009*) in saline solution (440 mM NaCl, 11 mM KCl, 26 mM MgCl$_2$, 13 mM CaCl$_2$, 11 mM Trizma base, 5 mM maleic acid, pH 7.45). In brief, the stomach was dissected from the animal. The intact stomatogastric nervous system (STNS) was isolated from the stomach, including: the two bilateral commissural ganglia, esophageal ganglion, and stomatogastric ganglion (STG), as well as the *lvn*, *mvn*, *dgn*. The STNS was pinned down in a Sylgard-coated petri dish (10 mL) and continuously superfused with chilled saline.

## Electrophysiology and dye-fills

The STG was desheathed and intracellular recordings from somata were performed with 20–30 MΩ glass microelectrodes filled with internal solution (10 mM MgCl$_2$, 400 mM potassium gluconate, 10 mM HEPES buffer, 15 mM NaSO$_4$, 20 mM NaCl as in *Hooper et al., 2015*). Intracellular signals were amplified with an Axoclamp 900A amplifier (Molecular Devices). For extracellular nerve recordings, Vaseline wells were built around the *lvn, mvn,* and *dgn* and stainless steel pin electrodes were used to monitor extracellular nerve activity (*Figure 1A*). Extracellular nerve recordings were amplified using model 3500 extracellular amplifiers (A-M Systems). Data were acquired using a Digidata 1440 digitizer (Axon Instruments) and pClamp data acquisition software (Axon Instruments, version 10.5). For GM identification, one of two electrodes was impaled into the soma and spiking activity was matched with GM spike units on the *dgn.* GM identity was verified with positive and negative current injections (*Figure 1D*). Following unambiguous identification, the GM soma was impaled with a second electrode containing dilute alexa488 dye (2 mM Alexa Fluor 488-hyrazide sodium salt (ThermoFisher Scientific, catalog no. A-10436, dissolved in internal solution)). The GM neuron was iontophoretically dye-filled with negative current pulses (−4 nA, 500 ms at 0.5 Hz) for 15–25 min. For two-electrode current clamp, the electrode containing alexa488 was typically used for recording and amplified with a 0.1xHS headstage. The electrode used for cell identification was used for current injection and amplified with a 1xHS headstage. Resting membrane potential and input resistance were monitored throughout the experiment to ensure the integrity of the preparation (neurons with input resistances <5 MΩ were discarded). In TTX, the mean input resistance, as measured at the somata and in the linear range of the current-voltage curve, was 9.9 ± 2.9 MΩ across preparations (*Table 2*). Input resistances did not change significantly before and after addition of TTX to the bath (data not shown). Reversal potentials for the glutamate response were determined by evoking at least three responses at ≥6 membrane potentials spanning −110 mV to −40 mV. In three preparations, an offset in the membrane voltage recording occurred during the dye-fill. This offset remained unchanged for the remainder of the experiment. This offset was corrected post-hoc during analysis of the recordings.

## Focal glutamate uncaging

For photo-uncaging experiments, preparations were superfused with a multi-channel Ecoline re-circulating pump (Ismatec/Harvard Apparatus, catalog no. PY2 72–6432) to maintain a stable bath volume and superfusion rate. 250 µM MNI-caged-L-glutamate (dissolved in saline; Tocris Bioscience, catalog no. 1490) was bath applied. $10^{-7}$ M teterodotoxin (TTX) was also superfused to minimize spike-driven synaptic activity. Alexa488-filled GM neurons were visualized with a custom-built epi-fluorescence microscope (*Figure 10*) equipped with a 40x water-immersion UV fluorescence objective (Olympus, LUMPLFLN 40XW) and a 470 nm LED (Thor Labs, M470L2). The emitted fluorescence was imaged with a monochrome CCD camera (Scientifica, SciCam). Focal photo-activation of MNI-glutamate was achieved with a small ultraviolet (UV) spot (~10 µm in diameter; *Figure 4—figure supplement 1*) projected through this same 40x objective lens. By situating the custom microscope on a micromanipulator (Sutter MPC-200), this UV spot could be lased at different positions on the GM neuronal structure. Three-dimensional coordinates for each photo-uncaging site were tabulated

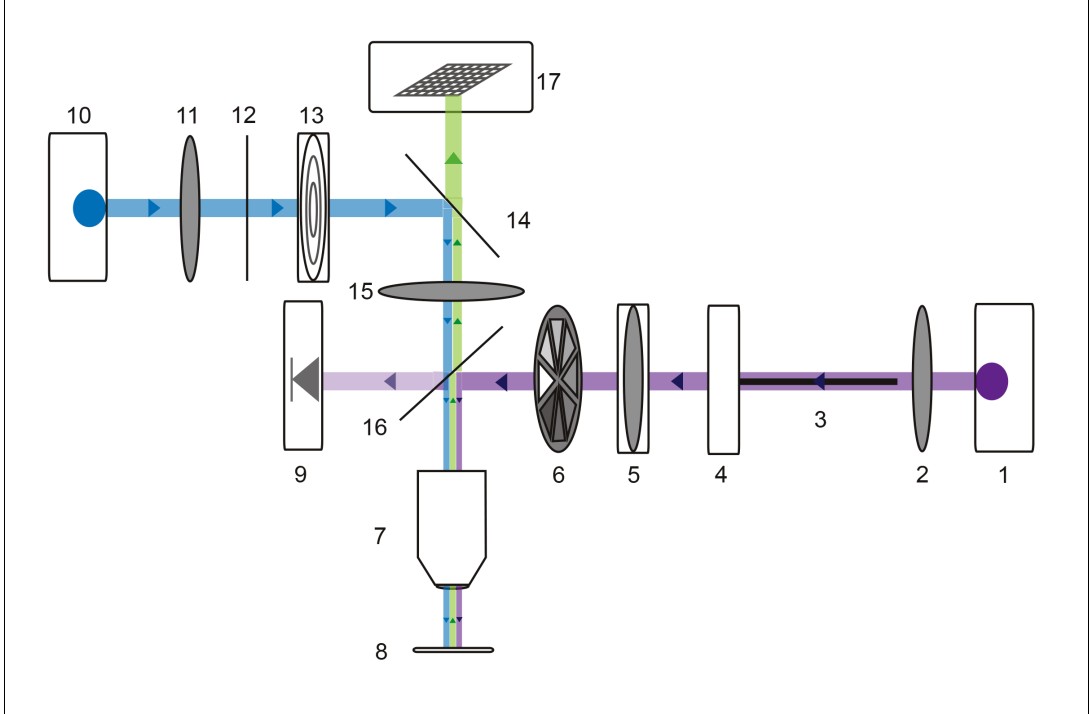

**Figure 10.** Microscope schematic showing laser (purple), fluorescence excitation (blue), and fluorescence emission (green) paths. A 1 Watt 355 nm laser (1) is focused by a plano-convex lens (2) and coupled to a 50-micron fiber optic cable (3) situated on an x-y translating fiber adaptor (4). The UV beam is collimated with a plano-convex lens situated in a z-translator (5). The beam intensity can be manually adjusted with a neutral density filter wheel (6) and passes through a 40x UV water immersion objective (7) onto the STNS preparation (8). The UV beam intensity is measured in real-time with a calibrated photo-diode (9) measuring a signal proportional to the power administered to the preparation. A 470 nm blue LED (10) is collimated by a plano-convex lens (11) and passes through a 466/40 nm band-pass filter (12) and field diaphragm (13). This light is re-directed by a 495 nm dichroic beamsplitter (14) and passes through a tube lens (15) and 405 nm low-pass dichroic beamsplitter (16). This is directed through the same 40x objective (7) and excites the alexa488 dye-filled preparation. This fluorescence passes through the 405 nm low-pass dichroic beamsplitter (16), tube lens (15), and 495 nm dichroic beamsplitter (14) and is subsequently captured by the CCD array of a firewire monochrome camera (Scientifica).

during the experiment. Glutamate responses were evoked with 30 mW (as measured at the back aperture of the objective), 1 ms UV pulses with inter-pulse periods no less than 30 s to minimize photo-damage and desensitization (*Figure 4—figure supplement 1*). To achieve this small UV spot, the UV laser beam (DPSS Lasers, model no. 35-07–100, 0.96 W, 100 kHz repetition rate) was coupled to a 50 µm diameter fiber optic cable (Thor Labs, M50L02S-A) with a UV lens (Thor Labs) and collimated with a 50 mm focal length plano-convex lens (Thor Labs, LA4148-UV). This collimated UV beam was delivered into the 40x objective lens to produce a focused spot of UV light on the preparation. A neutral density filter wheel was situated in the beam path for manipulation of the beam intensity (Thor Labs, NDM2). To ensure consistent beam intensities across experiments, the intensity was monitored in real-time using a photodiode (Thor Labs, PDA25K) previously calibrated with a power meter (Thor Labs, S302C). For precise temporal control of UV stimuli, a shutter (Thor Labs, SH1) was situated in the beam path. Both the laser Q-Switch and shutter were triggered by a set of coupled model 2100 isolated pulse stimulators (A-M Systems). The effective photo-uncaging radius in the x-y plane (15 µm) was determined by photo-uncaging at peripheral, distal neurites and moving away at 5 µm increments in the x-y plane (*Figure 4—figure supplement 1*) in a number of neuron types: pyloric dilator (PD), lateral pyloric (LP), and GM neurons.

## Dye-fill amplification and immunohistochemistry

Following photo-uncaging experiments, GM neurons were secondarily dye-filled with 2% Lucifer Yellow CH dipotassium salt (LY; Sigma, catalog no. L0144) in filtered water using a low-resistance electrode (10–15 MΩ). LY was injected for 20–50 min with negative current pulses (−6 to −8 nA, 500 ms

at 0.5 Hz). Once fine neurites of the cell could be visualized with a fluorescent stereomicroscope (Leica MF165 F), a preliminary image was acquired at 11.5x magnification with an attached mono-chrome digital camera (Leica DFC365 FC). LY-filled preparations were fixed for 40 min at 21°C or overnight at 4°C in 2% paraformaldehyde in phosphate-buffered saline (PBS; 440 mM NaCl, 11 mM KCl, 10 mM $Na_2HPO_4$, 2 mM $KH_2PO_4$, pH 7.4). Preparations were washed with 0.1 M PBS-T ((0.1–0.3%% Triton X-100 in PBS) and stored in PBS for 0–3 days prior to immunohistochemistry. The LY signal was amplified by 16 hr incubation with a polyclonal rabbit anti-LY antibody (1:500; Molecular Probes). After washing 5 × 15 min in PBS-T at room temperature, preparations were incubated in a secondary Alexa Fluor-488-conjugated goat-anti-rabbit antibody (1:500; Molecular Probes) for 1.5 hr at room temperature. Preparations were washed 5 × 15 min in PBS at room temperature before mounting on pre-cleaned slides (25 × 75 × 1 mm, superfrost, VWR) in Vectashield (Vector Laboratories, Burlingame, CA), with 9 mm diameter, 0.12 mm depth silicone seal spacers (Electron Microscopy Sciences, Hatfield, PA) under #1.5 coverslips (Fisher Scientific). Mounting in Vectashield with a spacer was sufficient to maintain the 3-dimensional structure of the neuron and ganglion (as in *Goeritz et al., 2013*).

## Confocal imaging and 3D reconstructions

Confocal stacks of the LY-filled neurons were acquired with a SP2 Leica Microscope and Leica Application Suite Advanced Fluorescence (LAS AF) software. Image stacks were acquired with a 20x dry objective (Leica HC PL APO CS 20x) at 1024 × 1024 resolution in 0.5 μm steps. Image stacks were visualized in both FIJI (ImageJ) software and KNOSSOS 3D image visualization and annotation software (developed by teams at the Heidelberg University and Karlsruhe Institute of Technology, employed by the Max Planck Institute for Medical Research, and freely distributed at: http://www. knossostool.org/). KNOSSOS software was used to manually trace and generate skeletons of the GM neuronal structures in three dimensions (*Supplementary file 1*-Neuronal Structures Hoc files; as in *Otopalik et al., 2017*). It is important to note that the resolution used here allowed us to reconstruct many neurons, but resulted in smaller total cable lengths and branch point numbers than reported in *Otopalik et al. (2017)*, where neuronal dye-fills were imaged at 60x magnification. This higher magnification would have precluded completion of the reconstructions and photo-uncaging experiments reported here.

## Morphological analysis

Following 3D skeleton generation, morphological analyses were completed across all GM neurons (n = 14), using a suite of custom analysis scripts written in Python using the iPython command line (freely available at: https://python.org and https://ipython.org, respectively) by AS. For each skeleton, branch points, lengths, and orders were measured in reference to the soma and used to generate dendrogram representations with normalized path lengths. Branch, or path, lengths were measured as the most direct neurite path from the soma to each branch tip. Tortuosities were calculated for each path length, as the ratio of the path length over the Euclidean distance from soma to branch tip. Axon locations were identified as the last branch points without terminating branch tips. Photo-uncaging positions were re-located on preliminary fluorescence images of the Lucifer yellow dye-fill (as situated during experiment, at 11.5x magnification) using a custom alignment script written in MATLAB (Mathworks, version 2015b) by AO, and then manually re-located in the 3D skeleton using KNOSSOS (*Supplementary file 2*-Uncaging Coordinates Hoc files). The branch order, path length, and diameter were determined for each photo-uncaging site, based on the confocal image stack. All quantitative morphology analysis scripts are freely available at the Marder Lab GitHub website (https://github.com/marderlab/Quantifying_Morphology).

## Electrophysiology analysis

Recordings acquired using Clampex software (pClamp Suite by Molecular Devices, version 10.5) and were visualized offline using a MATLAB waveform analysis toolbox written by Ted Brookings and analyzed with custom MATLAB scripts written by AO. Briefly, this pipeline of analysis scripts was used to detect and browse evoked glutamate responses, measure voltage response amplitudes and membrane potentials, plot raw recordings and processed data, and perform some statistical analyses. To determine reversal potentials at a given photo-uncaging position, raw response amplitudes

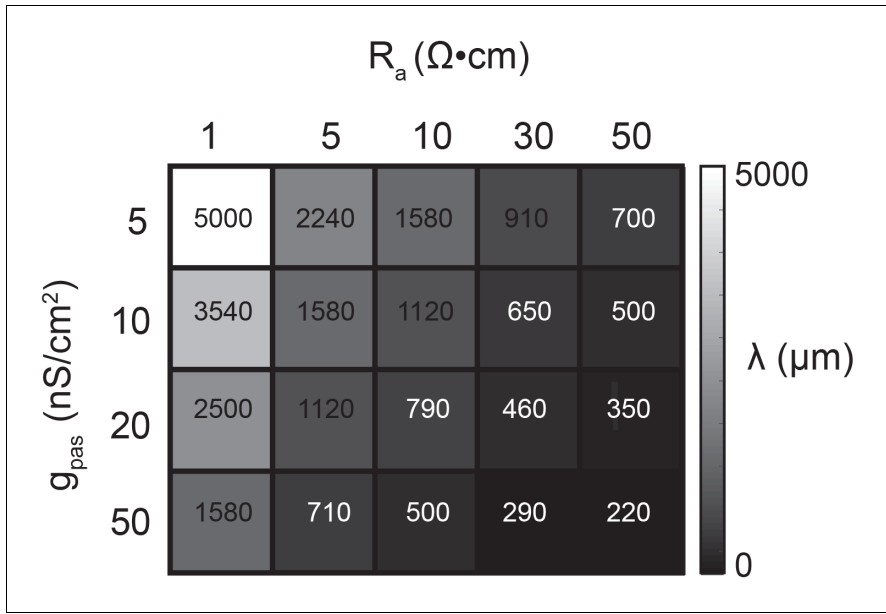

**Figure 11.** Passive parameters used for cable model library. All pairwise combinations. of passive conductance (gpas; 5, 10, 20, 50 nS/cm2) and axial resistance (Ra; 1, 5, 10,. 30, 50 Ω•cm) values were used to generate a library of 20 cable models with varying. electrotonic length constants (λ; shown numerically and in gray scale, in microns, as grid. elements below) between 220 μm and 5 mm.

were plotted as a function of membrane potential. These data were fit with a linear function (with an R value > 0.9 in all cases). In some cases, response amplitudes saturated and the linear regression analyses were performed only in the linear range of these data. The x-intercept of the resulting linear function indicated the reversal potential at that position. All electrophysiology analysis scripts are available at the Marder lab GitHub (https://github.com/marderlab).

## Passive cable models

A library of equivalent cylinder models was constructed in NEURON (freely available at: https://www.neuron.yale.edu/neuron), to simulate apparent reversal potential ($E_{rev}$) measurements of inhibitory responses generated at positions varying in distance from the recording site. The cables were uniform in their geometric properties: 1000 μm in length and 5 μm in diameter. All cables had a membrane capacitance of 1 μF·cm$^{-2}$. However, their passive properties were varied combinatorially with four passive conductance ($g_{pas}$) values (5, 10, 20, and 50 nS/cm$^2$) and five axial resistance ($R_a$) values (1, 5, 10, 30, 50 Ω·cm). Consequently, each cable can be distinguished by its electrotonic length constant (λ in μm) such that, $\lambda = \sqrt{\frac{rR_m}{2R_a}}$ where $R_m = \frac{1}{g_{pas}}$. λ values ranged between 200 μm and 5 mm (*Figure 11*). To measure the apparent $E_{rev}$s at the 0 μm end of the cable, the membrane potential was manipulated at the recording site with current injections between −8 and +2 nA. An inhibitory current (actual $E_{rev}$ = −70 mV, τ = 3 ms, $g_{max}$ = 1, 5, 10, or 50 nS) was activated at sites varying in distance from the recording site (at 0 μm). For a given cable and activation site, the apparent $E_{rev}$ was calculated by plotting the response amplitude (deltaV) as a function of membrane potential at the recording site. For each curve, linear regression analysis was completed (R > 0.9 and p<0.01 in all cases). The apparent $E_{rev}$s were identified as the x-intercept of the linear fit.

## Acknowledgements

We thank: Frank Mello for assistance in constructing mechanical components of rig; Bernardo Sabatini for optics expertise while constructing the custom microscope; Matthew Stenerson and Richard Ho for manual tracing of neuronal dye-fills; Philipp Rosenbaum for completion of several technical

experiments; Cosmo Guerini for generating additional analytical tools in Python; Edward Dougherty and the Confocal Imaging Lab at Brandeis University.

## Additional information

### Competing interests
EM: Deputy editor *eLife*. The other authors declare that no competing interests exist.

### Funding

| Funder | Grant reference number | Author |
|---|---|---|
| National Institute of Neurological Disorders and Stroke | F31NS092126 | Adriane G Otopalik |
| National Institute of Neurological Disorders and Stroke | R37NS017813 | Eve Marder |

The funders had no role in study design, data collection and interpretation, or the decision to submit the work for publication.

### Author contributions
AGO, Conceptualization, Data curation, Software, Formal analysis, Funding acquisition, Investigation, Visualization, Methodology, Writing—original draft, Writing—review and editing; ACS, Software, Writing—review and editing; MB, Resources, Supervision, Methodology; EM, Conceptualization, Resources, Supervision, Funding acquisition, Writing—original draft, Writing—review and editing

### Author ORCIDs
Adriane G Otopalik, http://orcid.org/0000-0002-3224-6502
Eve Marder, http://orcid.org/0000-0001-9632-5448

## Additional files

### Supplementary files
• Supplementary file 1. Neuronal Structure Hoc files.

• Supplementary file 2. Uncaging Coordinates Hoc files.

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
