## [Decision Letter]

[Editors’ note: a previous version of this study was rejected after peer review, but the authors submitted for reconsideration. The first decision letter after peer review is shown below.]

Thank you for submitting your work entitled "When complex neuronal structures may not matter" for consideration by *eLife*. Your article has been reviewed by three peer reviewers, and the evaluation has been overseen by a Reviewing Editor and a Senior Editor. The following individual involved in review of your submission has agreed to reveal their identity: Gilles Laurent (Reviewer #3). Our decision has been reached after extensive consultation between the reviewers. Based on these discussions and the individual reviews below, we regret to inform you that your work will not be considered for publication in *eLife* because we think addressing the issues will take longer than the allotted two months for revision. However, if you wish to undertake an extensive revision, we would be glad to reconsider the manuscript.

Summary:

The reviewers all found the topic interesting and commented on the quality of the experiments and the intriguing nature of the result. Nevertheless, they were mixed in their assessment of the manuscript. The strongest concerns were, specifically, that in the absence of a more detailed investigation of cable properties, the reported electrical compactness is not well or easily reconciled with the low input resistances, and generally, that some of the approaches might not be ideal (somatic recordings to infer dendritic properties, effects of glutamate uncaging with possibly non-uniform receptor density, synaptic reversal potentials to test compartmentalization, simplified model to explore dendritic complexity) and some key parameters might be inadequately constrained (electrical attributes of the dendrites, and effect of TTX on the measurements). The reviewers also expressed concern that without a resolution of the apparent conflict between the compactness and low input resistance, it is not be possible to judge whether or under what conditions the observations might apply to any other kinds of neurons. Addressing these concerns would involve (1) validating and/or justifying the uncaging approach, (2) providing information about passive cable properties, (3) reconsidering and/or discussing limits on synaptic reversal potentials, (4) extending modeling, e.g., to test whether passive cable properties might normalize PSP amplitude with distance; to more fully consider dendritic complexity; to explore the phenomenon with numerical simulation experiments that add plausibility arguments to the results. These points are explained in more detail in the reviewers' comments, which are included in full below.

*Reviewer #1:*

This study examines how variation in dendritic morphology influences synaptic integration in Gastric Mill (GM) neurons of the crab stomatogastric ganglion (STG). The authors combine somatic 2-electrode voltage-clamp recordings with focal glutamate uncaging experiments to show that the amplitudes of glutamate responses appear not to depend strongly on distance or variations in local dendritic geometry. These are interesting, experiments, and the questions addressed are of broad general interest. However, the paper suffers from the fact that the authors are constrained to infer dendritic properties from somatic recordings without any direct measurements, and there are assumptions about the nature of the glutamate uncaging that are not adequately justified. Also, the computer models do not explore the very dendritic complexity that the study purports to address. In the end, I don't think readers will be convinced that the current data set provides sufficient mechanistic insight into the nature of dendritic integration in GM neurons. My major comments are detailed below.

1) The premise of this study rests on the use of uncaged glutamate responses as a means to elicit uniform currents at many spatial locations in the dendrites. However, for this approach to be valid, the density of (presumably mostly extra-synaptic) glutamate receptors must be uniform across the dendritic arbor. While the uncaged responses appear quite reliable in Figure 4 at a given location, responses at adjacent locations in Figure 5 are highly variable, possibly reflecting variations in receptor density along the dendrites. There seems to be no systematic changes in response amplitude with local dendrite diameter or branch order, but it is not known what the amplitude of the responses are at the site of uncaging. With so many unknown free parameters (receptor density, surface area, local input resistance) it is not clear how these data can be interpreted cleanly. It is also not clear how these responses relate to the function of actual synapses, which according to the authors are located on the finer distal branches.

2) Considering how important passive cable properties are to the interpretation of the data, it is perhaps surprising how little information is provided about the actual passive electrical properties of GM cells. The only hint at these properties is the statement in the Methods that recorded GM cells had input resistances greater than 5 MΩ. This rather low value for input resistance (though perhaps not for invertebrate neurons) seems at odds with the authors' conclusion that GM neurons are electrically tight. There are modeling parameters reported in Table 2, but it is not clear whether these values are based on real measurements or just reasonable guesses. Some more detailed information about GM cell electrical properties would be helpful and important.

3) In general, synaptic reversal potentials are not sensitive predictors of the extent of dendritic compartmentalization. The manner in which reversal potentials are measured produces conditions that reduce the influence of dendritic filtering (since voltages during long command stimuli have reached a steady state). Also, voltages show less attenuation in the centrifugal vs. centripetal direction due to decreasing diameter and surface area of more distal dendritic branches. Williams and Mitchell (2008) elegantly explored these and other issues of dendritic filtering in neocortical pyramidal neurons…and yet these neurons show compartmentalized responses and significant attenuation of synaptic events propagating from the dendrites to the soma. The ability of a voltage clamp circuit to measure reversal potentials of distal synapses with reasonable accuracy does not necessarily mean that local and propagated synaptic responses do not depend on dendritic morphology.

4) The discussion in the last paragraph of the subsection “Electrotonic Structure in Circuit Context" contrasts the authors' present results and interpretations with results from hippocampal CA1 pyramidal neurons, where distance dependent attenuation of synaptic potentials is compensated via an increase in postsynaptic AMPA receptors (not voltage-gated conductances, as stated in the manuscript). In neocortical pyramidal neurons, though, distance dependent compensation for much of the dendritic voltage attenuation of synaptic events arises from the passive electrotonic structure of the dendritic arbor (e.g. Williams and Stuart 2002). It seems to me that either or both of these mechanisms could potentially explain the current findings. Given that neither of these mechanisms is explored directly in the current study, I don't know that the authors are in a strong position to argue that GM neurons operate differently.

*Reviewer #2:*

This manuscript reports an unusual electrotonic compactness in one neuron type in the crustacean stomatogastric ganglion. The GM neurons have varied morphology and dendritic branching patterns, yet all respond similarly during the motor pattern they participate in. The authors argue that this is because the neurons are very electrotonically compact, and current loss from distal dendrites is small. The experiments to show this are well done and convincing, including an interesting failure to detect a distance dependence of the apparent reversal potential of the synaptic responses. They do no experiments to determine how this compactness occurs, nor do they provide any explanation for this result. This discussion is important because to a naive reader the results appear to contradict simple cable property measures of current flow in branched processes. If such explanation (preferably with experimental verification) were provided, this would be a much stronger manuscript.

*Reviewer #3:*

This very nice paper explores the morphological and electrical geometry of an identified neuron (GM) in the crustacean STG. The authors find that the morphological features of this neuron are variable across animals, but that the responses, as measured from the soma, of the same neurons to uncaged Glu at various locations on its neuritic tree, vary little (across sites for a given neuron). Because most synaptic interactions are slow, and in a mostly passive regime, this leads to the conclusion that the electronic structure of GM is compact and its morphological architecture not relevant for integration/computation.

The paper is beautifully written, the figures limpid, the work quite extensive, the literature well researched and integrated. The main important result is that dendritic geometry may matter little when signals are slow and electrotonic structure compact.

[Editors’ note: what now follows is the decision letter after the authors submitted for further consideration.]

Thank you for submitting your article "When complex neuronal structures may not matter" for consideration by *eLife*. Your article has been reviewed by three peer reviewers, and the evaluation has been overseen by a Reviewing Editor and Gary Westbrook as the Senior Editor. The following individual involved in review of your submission has agreed to reveal his identity: Gilles Laurent (Reviewer #1).

The reviewers have discussed the reviews with one another and the Reviewing Editor has drafted this decision to help you prepare a revised submission.

Summary:

This work demonstrates that variation in morphology of the Gastric Mill (GM) neuron of the crab stomatogastric ganglion (STG) has relatively little influence on synaptic integration and the resultant voltage responses, owing to electrical compactness of the neurons.

Essential revisions:

The reviewers agreed that the appealed and revised version of the manuscript was improved, with many clarifications of the points that were initially raised. Two essential points remain:

1) The first, extensively discussed by the reviewers, has to do with whether the conclusion that morphology has relatively little effect on voltage responses is adequately supported by the modeling, given that the specific parameters of neurite morphology were not considered. Reviewers recognized that the main point may not be to rule out the idea that any dendritic computations took place, but rather that the slow graded signals characteristic of these cells are not greatly affected by morphology; nevertheless, it was acknowledged that this distinction could be emphasized further. As stated in the consultation, "The main thesis of the paper is that a long length constant of the neurite allows voltages from anywhere in the dendritic tree to propagate throughout the cell with minimal voltage attenuation, so the passive properties of their specific cell morphology is a central concern." One way to address this point, would be to import a real GM cell morphology into NEURON and express their passive properties uniformly in all compartments to test (a) whether the voltage attenuation or amplitude at the soma similar for long currents injected in distal neurites and (b) whether the length constant depends significantly on direction of propagation. The outcomes could serve either to support the results or place appropriate constraints on the conclusions. However, the reviewers agreed to leave the specific way of allaying these concerns to you, i.e., such a model is not required if you find alternative ways to clarify and/or limit the conclusions. The original "major comments" on this matter are included below for your reference, to guide and inform your revision.

2) The second point has to do with the placing constraints on the basis for electrotonic compactness. The Discussion mentions two possible explanations for how neurons with very low input resistance can be so electrotonically compact: very low internal resistance due to large diameter of initial processes (though this would not apply to the fine processes where the input-output synapses are located) and possibly high membrane resistance at branch points (though this would not block the passive spread of current along the internal resistance of the branches). The reviewers pointed out that it would be informative for the estimates of the length constant (1.5 mm) to be integrated with measurements of the input resistance (~ 10 MΩ) to come up with boundaries of Ri and Rm (given that Rin = (2/π)(RmRi)1/2(d)3/2 for a semi-infinite cylinder, and λ= ((RmRi)(d/4))1/2 (from Rall, 1977).

Comments related to Essential revision 1:

1) My main concern, before and now, has to do with whether the authors have truly shown that dendritic morphology plays little role in shaping voltage responses. The authors' argument is that the dendrites and soma are nearly equipotential, and that as a result inputs on any part of the structure have similar voltage contributions throughout the arbor. I don't see that this hypothesis has been adequately supported because the modeling does not take into account the actual neurite morphology the authors have quantified. The simulations in Figure 7 show that under conditions favorable for voltage propagation (a moderately large, constant diameter neurite exhibiting a long 800 µm length constant), there is still >50% attenuation of PSPs along its length despite the fact that the Erev can be accurately measured at these same distances. But the authors show in several figures (e.g. Figure 5) that there are striking reductions in diameter in the more distal regions, which will impart a directional asymmetry in the efficacy of voltage propagation. In a passive neuron, centrifugal propagation will be more effective in a tapering structure such as the one exhibited by GM neurons, which help explain why Erev can be measured so effectively even at distal uncaging sites in experiments. However, propagation toward the larger diameter neurites and soma will be comparatively unfavorable. The simulations in Figure 7 may thus underestimate the attenuation of voltages during propagation toward larger neurites.

I think the authors need to examine voltage propagation in a realistic morphological structure. If a conductance is introduced in neurites of differing orders, diameters and distances from the soma in a model neuron with realistic morphology, would these events yield comparable voltages at the soma? I do not expect that the authors must necessarily provide a full mechanistic explanation for their results, but I think such an examination would provide a better understanding of how spatially compact GM neurons are under more realistic conditions, and whether passive properties are sufficient to explain their uncaging results.

[In the words of another reviewer:] The other issue to discuss (also from Rall) is the large difference in voltage attenuation depending on the direction of the current flow (from a single dendritic point to the soma vs. from the soma to the dendrites). Your measurements of ipsp amplitude after stimulation at a single point are examples of the first, while your measurements of the Vrev from the soma are examples of the second. I think this will not be a problem for your analysis, but it should be explicitly discussed in the Discussion. The integration of spatially distributed synaptic inputs by the neuron would be an example of current flow from the periphery to the center, and from basic principles might show more attenuation.

2) There were some misunderstandings concerning my previous comments regarding the diversity of responses at adjacent locations and the lack of diversity of responses from neurites of different diameters. Put a different way, if a similar response is obtained from uncaging a 10 µm spot over a 1 vs. 10 µm diameter neurite (for example), does this not imply that there must be some other mechanism(s) in place to boost the amplitudes despite the 10-fold reduction in surface area and receptor density? A higher local input resistance might raise the local PSP amplitude, but such a mechanism might be limited due to the proximity of the reversal potential to rest. The authors have stated in their rebuttal that their concern is with voltage propagation and not local integration, but it seems to me that both issues are interrelated and central to understanding the results of their uncaging experiments.

3) In their rebuttal, the authors state that it is unlikely that receptor densities change along the dendrites because the "response amplitudes across sites within each neuron show no quantitative dependence on distance from the recording site." This assumes there is no significant centripetal attenuation of voltage, but as detailed earlier, there is some uncertainty in this premise. If there is even moderate voltage attenuation between the small diameter, distal neurites and the soma, would not another mechanism be needed to restore the amplitude of the response?

---

## [Author Response]

[Editors’ note: the author responses to the first round of peer review follow.]

*Summary:*

*The reviewers all found the topic interesting and commented on the quality of the experiments and the intriguing nature of the result. Nevertheless, they were mixed in their assessment of the manuscript. The strongest concerns were, specifically, that in the absence of a more detailed investigation of cable properties, the reported electrical compactness is not well or easily reconciled with the low input resistances, and generally, that some of the approaches might not be ideal (somatic recordings to infer dendritic properties, effects of glutamate uncaging with possibly non-uniform receptor density, synaptic reversal potentials to test compartmentalization, simplified model to explore dendritic complexity) and some key parameters might be inadequately constrained (electrical attributes of the dendrites, and effect of TTX on the measurements). The reviewers also expressed concern that without a resolution of the apparent conflict between the compactness and low input resistance, it is not be possible to judge whether or under what conditions the observations might apply to any other kinds of neurons. Addressing these concerns would involve (1) validating and/or justifying the uncaging approach, (2) providing information about passive cable properties, (3) reconsidering and/or discussing limits on synaptic reversal potentials, (4) extending modeling, e.g., to test whether passive cable properties might normalize PSP amplitude with distance; to more fully consider dendritic complexity; to explore the phenomenon with numerical simulation experiments that add plausibility arguments to the results. These points are explained in more detail in the reviewers' comments, which are included in full below.*

The low input resistances measured in these neurons was cause for concern by reviewers 1 and 2. We address this issue in the revised manuscript in several locations. We have listed the input resistances measured at the soma in TTX in Table 2; these measurements are described in the Methods subsection “Electrophysiology and Dye-Fills”, presented in the Results subsection “Distributed reversal potentials in GM neurons are nearly invariant”, and interpreted in the Discussion subsection “Physiological Implications”. Reconciling the low input resistances (as measured at the soma) with the electrotonic compactness of these structures may not be intuitive. In the Discussion, we write: “Given that these neurons are electrotonically compact, the input resistance as measured at the soma is likely a reflection of the resistance across the membrane surface area of much of the entire neuron. Thus, it is not surprising that input resistances measured at the soma are relatively low (mean of approximately 10 MΩ (Table 2), consistent with many years of recordings from STG neurons). If the neuron were more electrotonically compact, the input resistance measured at the soma would be higher, as the measurement would be restricted to the surface area of the local, somatic membrane.”

In the revised manuscript, we sought to clarify that the goal of this work is to assess how the passive cable properties of these neurons may influence voltage signal propagation. Thus, the term “compartmentalization,” is used very carefully and in reference to that which may be imposed by the passive cable properties of these neurons. It is true that there are different mechanisms (synaptic inputs, voltage-gated ion channels, modulatory receptors) that may superimpose on the passive cable structures and effectively compartmentalize these neurons in the intact circuit. But, probing these other mechanisms is not the objective of this work, but an intriguing phenomenon to address in future work. In the Discussion, we write: “These experiments were done in TTX, wherein TTX-sensitive, voltage-gated sodium channels are blocked and circuit activity is silenced. […] Future experiments in varying pharmacological and modulatory conditions could shed let on how different voltage- gated currents, modulatory currents, and ongoing synaptic input during rhythmic activity, may effectively compartmentalize these otherwise compact passive neuronal structures.”

In the new manuscript, we present an expanded passive cable model simulation (subsection “Probing electrotonic structure with reversal potential measurements”) that clearly shows:

i) the independence of apparent reversal potential measurements on receptor density (or maximal conductance).

ii) the dependence of apparent reversal potential measurements on activation site distance from the recording site

the contingency of this distance-dependency on the electrotonic length constant (λ) This simulation utilizes a library of cable models with varying membrane and axial resistances and electrotonic length constants varying between 200 µm and 5 mm. By expanding the parameter and stimulus spaces of the simulation, we believe we have shown these three above results and resolved much of the confusion expressed by reviewers 1 and 2.

Courses of Action

In the general summary, we were asked to address these issues by taking the following courses of action:

Validating and/or justifying the uncaging approach. In the Introduction, we write that previous work had used dual recordings in the soma and primary neurite to probe passive filtering of high and low frequency voltage signals. However, “relevant voltage events must arise at more distal, finer processes, where pre- and post-synaptic connections are located (King 1976a, b; Kilman and Marder, 1996). […] We present a surprising case wherein geometrical complexity and variability appear not to constrain passive physiology”. We feel that this is justification for use of this methodology, which provides a means of activating voltage events at sites varying in distance from the recording site. Given the unmistakable relationship between the apparent reversal potentials for responses evoked across the neuronal structure and the electrotonic length constant (as described in subsection “Probing electrotonic structure with reversal potential measurements”), this methodology is an adequate approach to address our scientific objective.

Providing information about passive cable properties. As described above, the input resistance measurements are now listed in Table 2, described in the Methods and Results, and interpreted in the Discussion. The geometrical properties are also described: neurite lengths are shown in Figure 2, diameters are presented in Figure 5 and Figure 8, and neurite taper in diameter is discussed in the subsection “Physiological Implications”.

*Reconsidering and/or discussing limits on synaptic reversal potentials and extending modeling.* Both of these points are addressed with the expanded passive cable simulations and are sufficient for addressing the clarified objective of this study.

*Reviewer #1:*

*This study examines how variation in dendritic morphology influences synaptic integration in Gastric Mill (GM) neurons of the crab stomatogastric ganglion (STG). The authors combine somatic 2-electrode voltage-clamp recordings with focal glutamate uncaging experiments to show that the amplitudes of glutamate responses appear not to depend strongly on distance or variations in local dendritic geometry. These are interesting, experiments, and the questions addressed are of broad general interest. However, the paper suffers from the fact that the authors are constrained to infer dendritic properties from somatic recordings without any direct measurements, and there are assumptions about the nature of the glutamate uncaging that are not adequately justified. Also, the computer models do not explore the very dendritic complexity that the study purports to address. In the end, I don't think readers will be convinced that the current data set provides sufficient mechanistic insight into the nature of dendritic integration in GM neurons. My major comments are detailed below.*

These experiments utilized two-electrode current clamp, not voltage clamp, at the soma to measure voltage responses evoked at varying distances from the somatic recording site. The objective of this work was not to address local dendritic integration; in the revised manuscript, we sought to clarify that the goal of this work is to assess how the passive cable properties of these neurons may influence voltage signal propagation. This concern has been addressed in response to the Summary above.

*1) The premise of this study rests on the use of uncaged glutamate responses as a means to elicit uniform currents at many spatial locations in the dendrites. However, for this approach to be valid, the density of (presumably mostly extra-synaptic) glutamate receptors must be uniform across the dendritic arbor. While the uncaged responses appear quite reliable in Figure 4 at a given location, responses at adjacent locations in Figure 5 are highly variable, possibly reflecting variations in receptor density along the dendrites. There seems to be no systematic changes in response amplitude with local dendrite diameter or branch order, but it is not known what the amplitude of the responses are at the site of uncaging. With so many unknown free parameters (receptor density, surface area, local input resistance) it is not clear how these data can be interpreted cleanly. It is also not clear how these responses relate to the function of actual synapses, which according to the authors are located on the finer distal branches.*

There were no efforts made to uniformly activate currents. The laser power, pulse duration, and glutamate concentrations were kept constant throughout the duration of the experiment (as described in the Results (subsection “Variable Glutamate Responses Across the Neuronal Structure”) and Methods (subsection “Focal Glutamate Uncaging”). This approach allowed us to measure the endogenous and heterogeneous glutamate sensitivities across different sites on a single neuron. The possibility of heterogeneous glutamate receptor densities as an explanation for the variable response amplitudes was discussed in the subsection “Variable Glutamate Responses Across the Neuronal Structure”, last paragraph, subsection “Distributed reversal potentials in GM neurons are nearly invariant”, first paragraph and subsection “Physiological Implications”, second paragraph. In the Discussion, we write: “Other neuron types compensate for passive attenuation of voltage responses with distance- dependent scaling of synaptic receptor density (Andrásfalvy and Magee, 2001; Magee and Cook, 2001; Smith et al., 1990). […] Because the response amplitudes across sites within each neuron show no quantitative dependence on distance from the recording site (Figure 5; Supplement to Figure 5; Table 1), it is unlikely that receptor densities are scaling with distance in a systematic way.” It is true that we cannot be certain of receptor densities without measuring responses at the photo-uncaging sites. Here, we wished to express that receptor densities appear not to scale so as to result in similar amplitudes after passive propagation to the somatic recording site (as has been shown in these other cited cases).

*2) Considering how important passive cable properties are to the interpretation of the data, it is perhaps surprising how little information is provided about the actual passive electrical properties of GM cells. The only hint at these properties is the statement in the Methods that recorded GM cells had input resistances greater than 5 MΩ. This rather low value for input resistance (though perhaps not for invertebrate neurons) seems at odds with the authors' conclusion that GM neurons are electrically tight. There are modeling parameters reported in Table 2, but it is not clear whether these values are based on real measurements or just reasonable guesses. Some more detailed information about GM cell electrical properties would be helpful and important.*

We have addressed this concern in response to the Summary above.

*3) In general, synaptic reversal potentials are not sensitive predictors of the extent of dendritic compartmentalization. The manner in which reversal potentials are measured produces conditions that reduce the influence of dendritic filtering (since voltages during long command stimuli have reached a steady state). Also, voltages show less attenuation in the centrifugal vs. centripetal direction due to decreasing diameter and surface area of more distal dendritic branches. Williams and Mitchell (2008) elegantly explored these and other issues of dendritic filtering in neocortical pyramidal neurons…and yet these neurons show compartmentalized responses and significant attenuation of synaptic events propagating from the dendrites to the soma. The ability of a voltage clamp circuit to measure reversal potentials of distal synapses with reasonable accuracy does not necessarily mean that local and propagated synaptic responses do not depend on dendritic morphology.*

Much of these concerns are addressed in response to the Summary above. It is true that voltage propagation may face different degrees of electrotonic decrement, depending on propagation direction. This question was not addressed in this work. Here, we measured voltage propagation in what might be considered the “centripetal” direction, although it is important to note that these neurons are not bipolar in structure, and that neurites serve as sites of both input and output (King 1976a, b). Our investigation of voltage propagation from neurite sites to the soma is relevant to the function of these neurons, given that synaptic events arise at secondary and higher-order neurites and must eventually propagate ‘centripetally’ toward the primary neurite and out to axonal projections to evoke spikes. As reviewer #1 stated, substantial decrement of voltage signals is seen in the centripetal direction in hippocampal neurons (Williams & Stuart, 2008). Our results are, indeed, a counter-example to what is seen in pyramidal neurons. We hope that readers will appreciate this result as a demonstration that not all neurons utilize the same strategies to produce reliable physiology, and that pyramidal neurons present one case in neuronal physiology, not the rule.

*4) The discussion in the last paragraph of the subsection “Electrotonic Structure in Circuit Context" contrasts the authors' present results and interpretations with results from hippocampal CA1 pyramidal neurons, where distance dependent attenuation of synaptic potentials is compensated via an increase in postsynaptic AMPA receptors (not voltage-gated conductances, as stated in the manuscript). In neocortical pyramidal neurons, though, distance dependent compensation for much of the dendritic voltage attenuation of synaptic events arises from the passive electrotonic structure of the dendritic arbor (e.g. Williams and Stuart 2002). It seems to me that either or both of these mechanisms could potentially explain the current findings. Given that neither of these mechanisms is explored directly in the current study, I don't know that the authors are in a strong position to argue that GM neurons operate differently.*

Our reference to distance-dependence scaling of AMPARs has been corrected for this error. We’ve addressed the remaining concerns in this comment in response to reviewer #1, point #1.

*Reviewer #2:*

*This manuscript reports an unusual electrotonic compactness in one neuron type in the crustacean stomatogastric ganglion. The GM neurons have varied morphology and dendritic branching patterns, yet all respond similarly during the motor pattern they participate in. The authors argue that this is because the neurons are very electrotonically compact, and current loss from distal dendrites is small. The experiments to show this are well done and convincing, including an interesting failure to detect a distance dependence of the apparent reversal potential of the synaptic responses. They do no experiments to determine how this compactness occurs, nor do they provide any explanation for this result. This discussion is important because to a naive reader the results appear to contradict simple cable property measures of current flow in branched processes. If such explanation (preferably with experimental verification) were provided, this would be a much stronger manuscript.*

We hope that we have adequately addressed the general concerns of reviewer 2 by: 1) adding a section to the Discussion titled “Physiological Implications,” which discusses possible mechanisms for electrotonic compactness and 2) expanding our cable model simulation to more clearly demonstrate the direct relationship between apparent reversal potential invariance, across activation sites varying in distance from the recording site, and electrotonic structure.

This more comprehensive simulation also serves as justification for our experimental approach.

[Editors' note: the author responses to the re-review follow.]

*Essential revisions:*

*The reviewers agreed that the appealed and revised version of the manuscript was improved, with many clarifications of the points that were initially raised. Two essential points remain:*

*1) The first, extensively discussed by the reviewers, has to do with whether the conclusion that morphology has relatively little effect on voltage responses is adequately supported by the modeling, given that the specific parameters of neurite morphology were not considered. Reviewers recognized that the main point may not be to rule out the idea that any dendritic computations took place, but rather that the slow graded signals characteristic of these cells are not greatly affected by morphology; nevertheless, it was acknowledged that this distinction could be emphasized further. As stated in the consultation, "The main thesis of the paper is that a long length constant of the neurite allows voltages from anywhere in the dendritic tree to propagate throughout the cell with minimal voltage attenuation, so the passive properties of their specific cell morphology is a central concern." One way to address this point, would be to import a real GM cell morphology into NEURON and express their passive properties uniformly in all compartments to test (a) whether the voltage attenuation or amplitude at the soma similar for long currents injected in distal neurites and (b) whether the length constant depends significantly on direction of propagation. The outcomes could serve either to support the results or place appropriate constraints on the conclusions. However, the reviewers agreed to leave the specific way of allaying these concerns to you, i.e., such a model is not required if you find alternative ways to clarify and/or limit the conclusions. The original "major comments" on this matter are included below for your reference, to guide and inform your revision.*

We have chosen not to pursue additional modeling of a realistic GM morphology in NEURON for three predominant reasons:

i) The first reason is scientific. One take-home message of this work is that similar physiology can arise from variable morphologies. Thus, it is not scientifically sound to upload one GM neuronal structure into NEURON as a means of elucidating ubiquitous biophysical mechanisms underlying their compact electrotonic structures, when it is evident that there are many solutions.

The purpose of the reduced cable model in this manuscript (as in the present and original submissions) is to provide a theoretical but intuitive explanation for the experimental approach. On the other hand, a model of a fully reconstructed neuron would strive to provide a mechanistic explanation of the result. Due the issue of multiple solutions, a model of one fully reconstructed neuron would not yield more insight than that of the discussion regarding biophysical mechanisms provided in the Discussion of the manuscript.

ii) The second reason is practical. We are not equipped to accurately model the full neuronal structure in NEURON due to three issues pertaining to the cross- sectional areas of STG neurites being ovular, rather than circular. (1) We do not have the data for the cross-sectional areas of all the neurites composing any one neuronal structure. This is partially because no reconstruction software, to our knowledge, allows manual tracers to assign non-circular cross-sections. And, the collection of these data, even if possible, would take months. To date, this is why we have chosen skeletal reconstructions for interrogation of STG neuronal morphology. (3) NEURON assumes circular cross-sections, making it impossible to accurately model these neurons. (3) We would have to “make up” data about the ion channel distributions over the extended cable.

iii) The final reason is of a philosophical nature. In the present study, the data provide an interesting story in their own right: GM morphology is complex and variable; this variability may be masked by compact electrotonic structures. The data stand on their own.

However, we have addressed the reviewers’ concerns by clarifying the text of the manuscript with the following:

i) In the Introduction we emphasize the arrangement of pre- and postsynaptic sites and spike initiation zones on the neurite tree and explain the physiological relevance of current flow direction in these neurons. This is important because it provides a rationale for our assay of electrotonic structure. We write:

“Synaptic transmission between neurons is predominantly graded, inhibitory cholinergic and glutamatergic transmission (Eisen and Marder, 1982; Marder and Eisen, 1984; Maynard and Walton, 1975; Graubard, et al., 1980; Manor et al., 1997, 1999). […] This juxtaposition of synaptic input and output suggests that current will flow in all directions across the neurite tree, centripetally and centrifugally, in the intact circuit, and allow for integration of voltage signals arising from disparate loci on the neurite tree, should the neuron be sufficiently electrotonically compact.”

ii) In the Results section, we specify the direction of current flow in both the cable model simulations and the physiological experiments. We write:

“In this simulation paradigm, current was injected at the recording site and flowed from the recording site to the stimulation site, changing the membrane potential at the distal site. […] Even so, the observations of sizeable voltage events at the recording site and reasonable reversal potentials are suggestive of a level of electrotonic compactness that is relevant to voltage signal propagation in either direction”

“Using two-electrode current clamp at the soma, we measured apparent E_rev_s of local inhibitory responses evoked by focal photo-uncaging of glutamate at positions varying in distance from the somatic recording site. In these experiments, current was injected at the somatic recording site and flowed centrifugally from the recording site to the photo- uncaging site, changing the membrane potential at this distal site.”

iii) In the Discussion, we revised our interpretation of the data and discuss how the result relates to the direction of the current injection and direction of initiated voltage signal propagation. We write:

“In GM neurons, synaptic voltage events may propagate tortuous neurite paths that extend beyond half a millimeter in length (Figure 2). […] In this sense, these neurons function almost like a single compartment, despite their complex structures.”

*2) The second point has to do with the placing constraints on the basis for electrotonic compactness. The Discussion mentions two possible explanations for how neurons with very low input resistance can be so electrotonically compact: very low internal resistance due to large diameter of initial processes (though this would not apply to the fine processes where the input-output synapses are located) and possibly high membrane resistance at branch points (though this would not block the passive spread of current along the internal resistance of the branches). The reviewers pointed out that it would be informative for the estimates of the length constant (1.5 mm) to be integrated with measurements of the input resistance (~ 10 MΩ) to come up with boundaries of Ri and Rm (given that Rin = (2/π)(RmRi)1/2(d)3/2 for a semi-infinite cylinder, and λ= ((RmRi)(d/4))1/2 (from Rall, 1977).*

As suggested, we did a series of calculations yielding values of R_i_ (Ω · cm) and R_m_ (Ω · cm^2^) for specified diameters (d), the observed average Rinput (10 MΩ), and predicted upper bound for λ (1.5 mm). For example, if d = 0.5 µm, R_i_ = 30 µΩ · cm and Rm = 550 µΩ · cm^2^. If d = 20 µm, R_i_ = 1.97 Ω · cm and R_m_= 0.88 Ω · cm^2^. We have chosen not to include these calculations because exact values for R_m_or R_i_would not do justice to the complex and variable structures observed across animals.

As a conceptual point, we estimated an effective λ (which is an upper bound) for the entirety of the neuronal structure. Presumably, the λ of specific segments of neurite may vary, depending on their biophysical and geometrical properties. As shown in Figure 6, neurite diameters (at the photo-uncaging sites) can range between 0.5 µm < d < 20 µm. In another anatomical study (Otopalik, et al., in review), we find that diameters of primary, secondary, tertiary, and terminating neurites can vary widely. While there is a trend of decreasing diameter from primary neurite to tip, a given neurite path does not necessarily taper linearly or exponentially with distance. In fact, diameters can increase and decrease non-systematically with distance (this can be observed visually in the confocal micrographs).

Given the wide range of diameters and λ values expected to arise across these neurite trees, one is left with an infinite matrix of possible combinations of resistivities and diameters that would yield the range of λ values sufficient for producing the compact neuronal physiology demonstrated in these experiments.

*Comments related to Essential revision 1:*

*1) My main concern, before and now, has to do with whether the authors have truly shown that dendritic morphology plays little role in shaping voltage responses. The authors' argument is that the dendrites and soma are nearly equipotential, and that as a result inputs on any part of the structure have similar voltage contributions throughout the arbor. I don't see that this hypothesis has been adequately supported because the modeling does not take into account the actual neurite morphology the authors have quantified. The simulations in Figure 7 show that under conditions favorable for voltage propagation (a moderately large, constant diameter neurite exhibiting a long 800 µm length constant), there is still >50% attenuation of PSPs along its length despite the fact that the Erev can be accurately measured at these same distances. But the authors show in several figures (e.g. Figure 5) that there are striking reductions in diameter in the more distal regions, which will impart a directional asymmetry in the efficacy of voltage propagation. In a passive neuron, centrifugal propagation will be more effective in a tapering structure such as the one exhibited by GM neurons, which help explain why Erev can be measured so effectively even at distal uncaging sites in experiments. However, propagation toward the larger diameter neurites and soma will be comparatively unfavorable. The simulations in Figure 7 may thus underestimate the attenuation of voltages during propagation toward larger neurites.*

*I think the authors need to examine voltage propagation in a realistic morphological structure. If a conductance is introduced in neurites of differing orders, diameters and distances from the soma in a model neuron with realistic morphology, would these events yield comparable voltages at the soma? I do not expect that the authors must necessarily provide a full mechanistic explanation for their results, but I think such an examination would provide a better understanding of how spatially compact GM neurons are under more realistic conditions, and whether passive properties are sufficient to explain their uncaging results.*

*[In the words of another reviewer:] The other issue to discuss (also from Rall) is the large difference in voltage attenuation depending on the direction of the current flow (from a single dendritic point to the soma vs. from the soma to the dendrites). Your measurements of ipsp amplitude after stimulation at a single point are examples of the first, while your measurements of the Vrev from the soma are examples of the second. I think this will not be a problem for your analysis, but it should be explicitly discussed in the Discussion. The integration of spatially distributed synaptic inputs by the neuron would be an example of current flow from the periphery to the center, and from basic principles might show more attenuation.*

*2) There were some misunderstandings concerning my previous comments regarding the diversity of responses at adjacent locations and the lack of diversity of responses from neurites of different diameters. Put a different way, if a similar response is obtained from uncaging a 10 µm spot over a 1 vs. 10 µm diameter neurite (for example), does this not imply that there must be some other mechanism(s) in place to boost the amplitudes despite the 10-fold reduction in surface area and receptor density? A higher local input resistance might raise the local PSP amplitude, but such a mechanism might be limited due to the proximity of the reversal potential to rest. The authors have stated in their rebuttal that their concern is with voltage propagation and not local integration, but it seems to me that both issues are interrelated and central to understanding the results of their uncaging experiments.*

These above concerns are addressed above in response to Essential revisions 1 and 2.

*3) In their rebuttal, the authors state that it is unlikely that receptor densities change along the dendrites because the "response amplitudes across sites within each neuron show no quantitative dependence on distance from the recording site." This assumes there is no significant centripetal attenuation of voltage, but as detailed earlier, there is some uncertainty in this premise. If there is even moderate voltage attenuation between the small diameter, distal neurites and the soma, would not another mechanism be needed to restore the amplitude of the response?*

In fact, we state that receptor densities across neurite sites likely do vary. But, our data suggests that they do not vary in a manner that is distance- or diameter-dependent. Because the reversal potentials are relatively constant across sites, we can rule out that the cable properties give rise to these variable response amplitudes. Thus, we are left with receptor density as the main determinant in voltage response amplitude heterogeneity. If receptor density scaled with distance or diameter (in such a way that compensated for these geometric properties) we might see a zero-slope in the linear fits, suggesting that voltage signal decrement is normalized by increasing receptor densities; or perhaps we would see increasing amplitude with distance, which might suggest over-compensation of receptor density; regardless, we would see some quantitative trend. To clarify our interpretation, we have re-written a section to read:

“Because the response amplitudes across sites vary in a manner that is independent of distance from the recording site (Figure 5; Figure 5—figure supplement 1; Table 1), it is unlikely that receptor densities are scaling with distance in a systematic way.”